# TaTToo: Tool-Grounded Thinking PRM for Test-Time Scaling in Tabular Reasoning

**Jiaru Zou**[1,2][*][†] **Soumya Roy**[2], **Vinay Kumar Verma**[2], **Ziyi Wang**[3][*] **David Wipf**[4][*],
**Pan Lu**[4], **Sumit Negi**[2], **James Zou**[4][†] **Jingrui He**[1][†]

[1]UIUC, [2]Amazon, [3]Purdue University, [4]The University of Hong Kong, [5]Stanford University

## Abstract

Process Reward Models (PRMs) have recently emerged as a powerful framework for enhancing the reasoning capabilities of large reasoning models (LRMs), particularly in the context of test-time scaling (TTS). However, their potential for supervising LRMs on tabular reasoning domains remains underexplored. Through detailed empirical analyses, we identify that existing PRMs, though widely adopted for supervising text-only reasoning steps, struggle with table-specific operations such as sub-table retrieval and schema interaction, leading to critical performance bottlenecks. To address this limitation, we propose TaTToo, a novel table-grounded PRM framework that (i) reasons explicitly over tabular reasoning steps and (ii) integrates tool-based verification to provide precise reward supervision. Concretely, we first design a scalable data curation pipeline that constructs over 60k high-quality step-level annotations by integrating table verification rationales with tool-based executions. Building on the collected data, we train TaTToo with a dual-stage paradigm: cold-start supervised fine-tuning to capture tool-use reasoning patterns, followed by reinforcement learning with tool-grounded reward shaping to align our model with table-based verification. We provide a comprehensive evaluation of the policy improvement induced by our newly designed PRM. Across 5 challenging tabular reasoning benchmarks covering numerical reasoning, fact-checking, and data analysis, TaTToo improves downstream policy LRMs by 30.9% at inference, surpasses strong PRM baselines such as Qwen-2.5-Math-PRM-72B with only 8B parameters, and demonstrates strong generalizability across diverse TTS strategies.

Figure 1: **Overview of TaTToo framework.** We first curate 60k high-quality instances by collecting expert verification rationales with tool integration (Section 4.2). We then train our PRM through a dual-stage training paradigm to achieve tool-grounded step-by-step reward supervision (Section 4.3).

---

[*]Work done while at Amazon

[†]Correspond to {jiaruz2,jingrui}@illinois.edu, jamesz@stanford.edu

## 1 INTRODUCTION

Tabular reasoning has become a fundamental capability for emerging large reasoning models (LRMs) across various real-world applications, including numerical analysis (Akhtar et al., 2023), fact-checking (Chen et al., 2019; Parikh et al., 2020), and question answering (Vakulenko and Savenkov, 2017; Li et al., 2023a). Unlike free-form text, tables encode information in rows and columns with an implicit relational semi-structure. Effective reasoning over tables therefore requires both accurate interpretation of tabular content and step-by-step logical inference to produce precise answers (Wang et al., 2024c; Zhang et al., 2025a). To support such multi-step reasoning, recent studies such as Table-R1 series (Wu et al., 2025b; Yang et al., 2025b; Jin et al., 2025) have incorporated reinforcement learning (RL) techniques (Schulman et al., 2017; Shao et al., 2024) to better align LRMs with the demands of complex table understanding and reasoning.

On the other hand, process reward models (PRMs) (Setlur et al., 2024; Wang et al., 2024b; Yang et al., 2024) have been developed to provide step-level supervision over model reasoning trajectories during test-time scaling (TTS), offering fine-grained verification that enhances LRMs' performance at inference. However, despite growing computational budgets and increasing emphasis on advancing LRMs' tabular reasoning abilities (Ye et al., 2025; Muennighoff et al., 2025), a corresponding step-level PRM to supervise the reasoning quality of these models in table domains is equally important but remains notably absent. This gap motivates our study of a fundamental question:

> 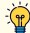 *How can we provide reliable step-level supervision to advanced LRMs in tabular reasoning?*

To investigate this question, we first revisit several general-domain advanced PRMs and evaluate their effectiveness in supervising table-involved reasoning steps generated by LRMs. Our analysis reveals that existing PRMs struggle to reliably verify two critical types of tabular CoT steps: ① *Table Retrieval*, where PRMs fail to supervise whether LRMs extract the correct sub-region of the input table relevant to the query; and ② *Schema Interaction*, where PRMs cannot detect attention collapse (Dong et al., 2021), as LRMs often overlook long-range table dependencies due to inherent locality bias. Beyond challenges arising from the tabular input modality, we also observe that current PRMs frequently introduce supervision errors within their own evaluation process, stemming from inaccurate table lookups or failed operations on tables. These shortcomings amplify bias and noise during TTS, ultimately creating persistent performance bottlenecks.

Motivated by our preliminary analyses, we propose **TATTOO**, a new **Ta**ble **T**hinking PRM with **Too**l integration abilities to provide more reliable and precise supervision for tabular reasoning. Distinct from prior PRMs that provide weak supervision over table-specific operations, TATTOO provides step-level supervision tailored to different input steps, applying both table-grounded rewards for tabular operation steps and inner-reasoning rewards for text-based reasoning steps. In addition, TATTOO can leverage several external tools to interact with table contents, execute code-based operations, and incorporate the results back into the step-by-step verification process. To build TATTOO, we first design a scalable data curation pipeline that yields over 60k high-quality supervision instances by integrating expert verification rationales with tool-based executions. We then train our PRM under a dual-stage paradigm: supervised fine-tuning to capture step-level tool-use reasoning patterns, followed by reinforcement learning with a newly designed reward shaping scheme to encourage effective tool manipulation and faithful reasoning for accurate verification. Finally, we provide theoretical intuition on the policy improvement induced by incorporating TATTOO during inference.

To demonstrate the effectiveness of TATTOO, we conduct extensive experiments on five challenging tabular reasoning benchmarks, covering table-based question answering, numerical reasoning, fact-checking, and data analysis. Across all benchmarks, incorporating 8B-size TATTOO improves downstream policy models by 30.9%. In addition, TATTOO consistently outperforms strong PRM baselines such as Qwen-2.5-Math-PRM-72B (Zhang et al., 2025b) and GenPRM-32B (Zhao et al., 2025) with up to 9x parameter efficiency. In-depth analyses further demonstrate that incorporating our dual-stage training paradigm yields a 10.2% improvement over standard PRM training, and TATTOO exhibits strong generalizability across diverse TTS strategies, including Beam Search and DVTS.

## 2 PRELIMINARY

**Table Understanding with LRMs.** We denote $T = (H, R)$ as a semi-structured table, where $H$ is the set of column headers defining the schema-level semantics, and $R$ is the set of rows, with each row

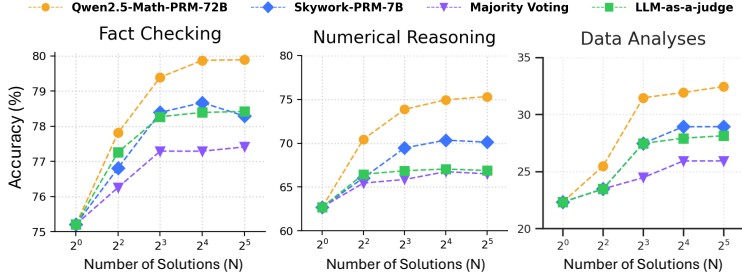
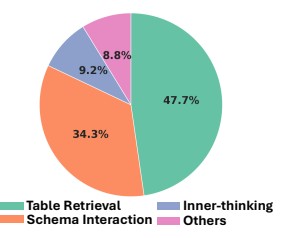

Figure 3: Error Distribution over 4 step categories across 500 incorrect cases after Best-of-N selection.

Figure 2: Best-of-N performance of DeepSeek-R1-Distill-Qwen-14B across 3 table tasks on TableBench with different types of step verifiers.

composed of cell entries aligned with $H$. Given a table $T$ and an associated natural language query $q$, we define a reasoning model as a conditional generation policy $\pi(\tau \mid T, q)$, where $\tau = \{a_1, \ldots, a_L\}$. Here, $\tau$ denotes the reasoning model's generated reasoning trajectory, including both intermediate reasoning steps $\{a_i\}_{i=1}^{L-1}$ and the final answer $a_L$. In our problem setup, the intermediate reasoning steps consist of both model inner-thinking reasoning traces and tool-integrated programs that operate directly on the table to retrieve or compute intermediate results. The final answer can take different formats depending on the query type, including textual or numerical values, boolean outputs (e.g., True/False), or executable programs (e.g., Python, SQL).

**Reward Modeling for Tabular Reasoning.** Given a table $T$, a query $q$, and a candidate response $\tau$ generated by a policy LRM, a standard step-level verifier (i.e., PRM) parameterized by $\theta$ computes a scoring function $\mathcal{R}_\theta(\cdot)$ that assigns step-level rewards $r_i$ evaluating the correctness of each step $a_i \in \tau$. The trajectory-level reward $r_\tau$ for each response $\tau$ is then obtained by aggregating these step-level rewards. Formally, we have:

$$r_i = \mathcal{R}_\theta(a_i \mid T, q, \tau_{<i}), \quad \text{with } r_\tau = \mathcal{A}(r_1, r_2, \cdots, r_L), \tag{1}$$

where $\mathcal{A}(\cdot)$ denotes an aggregation function such as MEAN and SUM (Liu et al., 2025). The rewards provided by the PRM $\mathcal{R}_\theta$ can be further leveraged by a test-time compute strategy $\phi$ (e.g., Best-of-N (Brown et al., 2024), Beam Search (Snell et al., 2024)) to guide resampling, refinement, and candidate selection among the responses generated by the policy model.

## 3 WHY TABLE REASONING REQUIRES VERIFIERS BEYOND CURRENT PRMS?

We begin by revisiting existing general-domain PRM methods to assess their effectiveness in supervising LRMs on tabular reasoning tasks and to identify potential performance bottlenecks. To this end, we conduct a pilot study guided by two key questions:

> ***RQ1 -*** Beyond free-form text inputs, can common general-domain PRMs combined with TTS strategies also enhance the performance of LRMs on tabular reasoning tasks?
>
> ***RQ2 -*** When step-level reward supervision is crucial for tabular reasoning performance, how can PRMs effectively supervise and guide the quality of each reasoning step generated by LRMs?

For brevity, we defer detailed experimental setups to Appendix F. To investigate RQ1, we evaluate various step-level verification methods, including two advanced PRMs (Qwen2.5-Math-PRM-72B (Zhang et al., 2025b) and Skywork-PRM-7B (He et al., 2024a)), majority voting (Liu et al., 2025), and LLM-as-a-judge (Zheng et al., 2023) with the Best-of-N TTS strategy. We choose DeepSeek-R1-Distill-Qwen-14B (Guo et al., 2025) as the common LRM and evaluate on TableBench (Wu et al., 2024), which includes three fundamental table tasks (Fact Checking, Numerical Reasoning, and Data Analysis). As shown in Figure 2, we observe that for small values of $N$, incorporating step-level verifiers into Best-of-$N$ generally improves LRM's performance over single-shot generation, with PRMs providing the largest gains. However, once the number surpasses a threshold ($N \geq 8$), accuracy across all three table tasks converges to a bottleneck. For example, the performance of Qwen2.5-Math-PRM-72B on fact-checking is 79.19%, 79.82%, and 79.84% for $N = \{8, 16, 32\}$, indicating that further increases in $N$ yield negligible gains, even though with PRM incorporation.

Table 1: Representative error cases in 3 different reasoning step categories. Each example highlights the erroneous model step in red, the corresponding error description, and the PRM's (mis)judgment reward, illustrating where existing PRMs fail to detect mistakes.

| Category | Wrong Model Answer | Error Description & PRM output |
|---|---|---|
| Table Retrieval | \<think\> Step 1: I need to figure out the average prominence of mountain peaks in the Democratic Republic of the Congo that have an elevation of at least 3000 meters. Step 2: First, I'll review the provided table. My main focus is on the 'country' and 'elevation (m)' columns. *(Wrong Step)* ...\</think\> Final Answer is \boxed{2306.65} | The model initially retrieves the wrong part of the table (columns *'Country'* and *'Elevation (m)'*), which leads to errors in all subsequent reasoning steps. **PRM → {correct}** ✗ Fail to detect the error |
| Schema Interaction | \<think\> Step 1: Let me start by looking at the table provided ...The retrieved columns are ... Step 4: I'll go through each row and note down the episodes: Episode 1: Rating - 5.1, Episode 2: Rating - 5.2 ...*(Wrong Step)* Step 5: Adding the rating together is 5.1 + 5.2 + ...\</think\> Final Answer is: \boxed{11.52} | The model identifies the correct column for the answer, but fails to extract all elements, omitting the last cell. This omission leads to incorrect reasoning in all subsequent steps. **PRM → {correct}** ✗ Fail to detect the error |
| Inner-thinking | \<think\> Step 1: Based on the table contents, I need to count podiums per season. 2008: $(1 + 1 + 1)/3 = 1$, 2009: 6, 2010: 0, 2011: $(2 + 0)/2 = 1$ ... Step 4: Then I will average across seasons. Final average is $(1 + 6 + 0 + 1)/4 = 2$ *(Wrong Step)* \</think\> Final Answer is: \boxed{2} | The model incorrectly does the calculation by averaging the season-level means, giving each season equal weight, instead of averaging across all team-seasons. **PRM → {incorrect}** ✓ Detect the error |

Figure 4: **Left:** PRM's rewards on 500 reasoning steps with the real-retrieved/randomly-replaced sub-table. **Middle:** Layer-wise average attention mass vs. relative step distance in tabular reasoning. Attention concentrates on nearby steps, with sharp decay as distance increases. **Right:** Best-of-N results on DeepSeek-R1-Distill-Qwen-14B for numerical reasoning with/without the table prefix.

> **Observation 1 (Limitation on TTS):** Existing PRMs yield modest improvements on tabular reasoning, but their efficacy quickly saturates, failing to fully exploit additional test-time compute.

**Error Analysis.** Building on the observation, we further investigate the underlying causes of the performance bottleneck by conducting an error analysis on LRM's generation and PRM's supervision processes. Specifically, we sample 500 erroneous Best-of-N responses (N= 32) selected by the PRM from LRM outputs, and ask human experts to classify them into 13 well-defined tabular error types (see Appendix C). We then connect these errors with 4 reasoning-step categories reflecting the typical flow of LRMs' reasoning process: (i) *Table Retrieval Steps*, locating relevant rows/columns regarding the input query; (ii) *Schema Interaction Steps*, reasoning over the retrieved table contents, (iii) *Inner-thinking Steps*, models' inner reasoning independent of table contents, and (iv) *Others*, initial setup or final output steps that are irrelevant to core reasoning process. Figure 3 presents the error distribution across 4 reasoning step categories. We find that most errors arise in *Table Retrieval* (47.7%) and *Schema Interaction* (34.3%), implying that PRMs perform reasonably well on independent reasoning but fall short when reasoning steps involve table-specific operations. For better demonstration, we provide representative examples for each category in Table 1.

**Why do PRMs fail on table-involved reasoning steps?** Next, we take a closer look at why PRMs lose their supervisory effectiveness when reasoning steps involve table operations. For *Table Retrieval Steps*, we conduct a contrastive experiment focusing particularly on the table contents retrieved by LRMs within their responses. We randomly sampled 500 responses and constructed two variants by (i) retaining the original LRM-retrieved sub-table, and (ii) replacing it with a randomly selected sub-table

region from the original input table. Figure 4 (left) shows the output rewards of Qwen2.5-Math-PRM-72B on both variants. The nearly identical distributions between real and random sub-tables indicate that current PRMs fail to distinguish retrieval correctness, suggesting that they are unable to assess whether the LRMs' retrieved portion of the table corresponds to the query.

> **Takeaway 1 (Table Retrieval):** Existing PRMs are insensitive to table retrieval correctness in the reasoning steps and fail to recognize whether the retrieved content corresponds to the query.

For *Schema Interaction Steps*, we found in prior experiments that in the logic flow of LRMs' trajectories, table retrieval steps typically occur at the beginning, as the model must first extract relevant information from the table to answer the query. In contrast, schema interaction steps frequently occur far sentences from the beginning table retrieval steps, since LRMs tend to perform intermediate reasoning before revisiting their retrieved contents when needed. Figure 4 (middle) illustrates the attention distribution of the LRM between the schema interaction step (step 8) and the table retrieval step (step 0). Due to the auto-regressive nature of LRMs, the schema interaction step attends primarily to nearby steps while assigning little attention to the earlier retrieval step. This inherent locality bias causes the model to frequently misinterpret or discard previously retrieved contents, even when the retrieval step has already extracted the correct information. Moreover, current PRMs fail to supervise such misinterpretations, as their evaluations are highly localized to the current step rather than capturing dependencies on distant prior steps (Zou et al., 2025b; Feng et al., 2025b).

> **Takeaway 2 (Schema Interaction):** Schema interaction steps under-attend to distant table retrieval contents due to locality bias. PRMs miss these failures since they can't look ahead and capture long-range dependencies among distant steps.

**Table Prefix is the Key.** To explore potential solutions to the limitation above, we begin with a simple input modification for PRMs: prepending the retrieved table contents as a prefix to each schema interaction step. This grants PRMs direct access to the retrieval context, alleviating the need for long-range dependencies. We evaluate this modification and report the results in Figure 4 (right). Incorporating the table prefix indeed improves PRM supervision and leads to stronger downstream LRM performance. However, directly applying the prefix remains challenging, as current PRMs cannot automatically identify schema interaction steps, and the table prefixes obtained from LRMs are not guaranteed to be correct without proper supervision.

**Motivation for TATTOO.** Our analyses above highlight the need for a principled step-level verifier capable of providing robust supervision over both table-grounded operations and models' inner reasoning. Motivated by this, we propose a new process reward model specifically designed to support LRMs in tabular reasoning.

## 4    BUILDING A TABLE-GROUNDED STEP VERIFIER

We introduce TATTOO, a generative PRM that provides reward supervision over both table operations and model inner thinking steps. Our method builds on two key components: (i) a large-scale data curation pipeline that synthesizes reasoning and tool usage for PRM training, and (ii) a dual-stage training paradigm that learns step-level verification with tool use optimization.

### 4.1    TABLE-AWARE AND TOOL-INTEGRATED SUPERVISION

**Table-Aware Reward.** To align with the LRM's reasoning process on table tasks, we separate the supervision of table operations from model's inner reasoning part and decompose TATTOO's step-level reward (Eq. 1) into two components:

$$r_i = \begin{cases} r_{i,\text{rea}}, & \text{if } a_i \in \text{inner-thinking}, \\ r_{i,\text{tab}}, & \text{if } a_i \in \text{table retrieval or schema interaction}, \end{cases} \quad \text{and } r_\tau = \frac{1}{L} \sum_{i=1}^{L} r_i, \quad (2)$$

where $r_{i,\text{rea}}$ captures the correctness of the model inner-reasoning process, $r_{i,\text{tab}}$ reflects the accuracy of table-grounded operations, and $r_\tau$ denotes the trajectory-level reward.

**Tool Integration in Verification.** A major limitation of current PRMs is their inability to supervise table-involved reasoning steps (as shown in Section 3). Meanwhile, recent studies (Feng et al.,

2025a; Qian et al., 2025) have shown that LLM agents can autonomously use **tools** to interact with external environments and iteratively refine their reasoning. In a similar spirit to address current PRM's limitation, we incorporate several external table-oriented tools into TᴀTTᴏᴏ's verification process to enable more reliable step supervision. We next describe how we curate a training set with tool-augmented, table-aware rewards and use it to train TᴀTTᴏᴏ.

## 4.2 TᴀTTᴏᴏ Data Curation Pipline

We design a large-scale data curation pipeline that simulates real-world scenarios of PRM tool use and step verification at scale. As illustrated in Figure 1, there are three main stages:

❶ **Reasoning Trajectory Generation.** We begin by collecting trajectory responses from expert LRMs (e.g., DeepSeek-R1 (Guo et al., 2025) and Claude-Opus-4.1 (Anthropic, 2025)) on table-based questions drawn from diverse benchmarks, including TableInstruct (Wu et al., 2024), HybridQA (Chen et al., 2020), ToTTo (Parikh et al., 2020), and WikiTQ (Pasupat and Liang, 2015b). We generate multiple responses per query and apply dual verification with human annotators and expert LLMs to filter out low-quality data, yielding a high-quality trajectory pool $\mathcal{T}_{\text{pool}}$ for subsequent labeling.

❷ **Verification Synthesis & Reward Assignment.** We next provide step-level verification rationales and reward labels for each candidate response in $\mathcal{T}_{\text{pool}}$. (i) For *table retrieval steps*, we extract the sub-table in each step and use LLM-as-a-judge to assess its relevance to the query, assigning table reward $r_{i,\text{tab}} \in \{-1, 1\}$ based on retrieval correctness. (ii) For *schema interaction steps*, we prepend the accurate sub-table as a table prefix to each collected verification rationale (according to our table-prefix analysis in Section 3) and assign $r_{i,\text{tab}} \in \{-1, 1\}$ based on the correctness of the specific table-based operations or reasoning. (iii) For *inner-thinking steps*, which involve no table contents, we apply LLM-as-a-judge and follow established labeling strategies (Zhao et al., 2025; Khalifa et al., 2025) to assign $r_{i,\text{rea}} \in \{-1, 1\}$ based on reasoning quality.

❸ **Tool Use Synthesis.** To train TᴀTTᴏᴏ to leverage tools for more accurate verification, we further augment the collected verification rationales with tool invocations, execution results, and feedback at the step level. Specifically, inside the rationale contents, we replace manual reasoning for table lookups or calculations with the corresponding tool call and its execution output. We primarily employ two types of table tools: (i) *Computation tools:* code snippets (e.g., Python, SQL) for arithmetic and aggregation over table inputs; (ii) *Table Lookup tools:* DataFrame APIs (e.g., Polars) or Lookup Utilities (e.g., CSV/Excel readers) for retrieving specific rows, columns, or cells during verification.

Finally, we construct over 60k high-quality training instances with complete verification rationales and step-level rewards. This dataset is then used to train TᴀTTᴏᴏ to integrate tool use with reasoning for robust step supervision. We leave additional data curation details in Appendix D.

## 4.3 Tool-Grounded Dual-Stage Training

With the training data recipe in place, we train TᴀTTᴏᴏ via a dual-stage paradigm: supervised fine-tuning to capture tool-integrated verification patterns, followed by RL-based policy optimization with a newly designed reward shaping scheme to further refine our PRM's step-level rationales and ensure accurate verification.

**Table-Aware Verification with Tools via SFT.** We first finetune our PRM $\mathcal{R}_\theta$ on the curated dataset (Section 4.2). Specifically, given a training instance $(T, q, \tau)$ consisting of a table $T$, a query $q$, and an LRM-generated trajectory $\tau = (a_1, \ldots, a_L)$, we train the PRM to output, for each step $a_i \in \tau$, a verification rationale $v_i$ together with its corresponding step-level reward $r_i$. By formulating PRM training as language modeling, $\mathcal{R}_\theta$ is optimized auto-regressively to (i) identify accurate sub-table regions, (ii) learn to dynamically incorporate the retrieved table prefix into each schema interaction step, and (iii) generate verification rationales with tool-integration patterns.

**Tool-Grounded Reward Shaping in RL.** Prior generative PRM approaches (Liu et al., 2025; Khalifa et al., 2025; Zhao et al., 2025) typically conclude PRM training after the SFT stage. In contrast, we draw inspiration from recent advances in agentic RL (Jaech et al., 2024; Guo et al., 2025; Zou et al., 2025c) and further apply policy optimization to more tightly align the PRM's verification process with effective tool utilization. Specifically, we optimize $\mathcal{R}_\theta$ with a modified GRPO (Shao et al., 2024) by providing dense, tool-grounded supervision signals during policy optimization. During RL rollouts of each training instance $(T, q, \tau)$, we replace the original rule-based GRPO supervision

signal with a denser per-step reward signal $s_i$, defined as:

$$s_i = \underbrace{\mathbb{1}\{\hat{r}_i = r_i\}}_{\text{label-matching}} - \underbrace{\lambda_{\text{cal}}\Big(-\log \mathcal{R}_\theta(r_i \mid T, q, \tau)\Big)}_{\text{confidence calibration}} + \underbrace{\lambda_{\text{tool}} \cdot \text{support}(\hat{v}_i)}_{\text{tool-grounding}}, \quad (3)$$

where $\hat{r}_i$ is the PRM's predicted step-reward and $r_i$ is the ground-truth step-reward for the input step $a_i \in \tau$; $\hat{v}_i$ denotes the verification rationale generated by the PRM at step $i$, and $\text{support}(\hat{v}_i) \in \{0, 1\}$ measures whether the rationale correctly incorporates tool outputs; and $\lambda_{\text{cal}}, \lambda_{\text{tool}}$ are tunable coefficients. Besides enforcing correctness with the *label-matching term*, the *confidence calibration term* stabilizes training by encouraging higher probability on the ground-truth label, and the *tool-grounding term* encourages rationales that effectively incorporate tool outputs. We then aggregate the per-step signals $s_i$ into a trajectory-level training reward, normalize it within each sampled group to compute group-relative advantages, and update the PRM $\mathcal{R}_\theta$ under the GRPO objective.

## 4.4 INFERENCE-TIME POLICY IMPROVEMENT – AN INTUITIVE VIEW

To intuitively elucidate the role of TATTOO and its table-aware rewards on LRM's tabular reasoning process (Eq. 2), we provide a theoretical analysis on the policy improvement induced by TATTOO.

Recall that the goal of our PRM is to improve the generated trajectory $\tau$ sampled from a policy LRM $\pi$, i.e., $\tau \sim \pi(\cdot \mid T, q)$. By combining the input table and query, we represent $(T, q, a_1, \ldots, a_{i-1})$ as the current state $\mathbf{s}_i$. At step $i$, the policy LRM $\pi$ samples an action $a_i \sim \pi(\cdot \mid \mathbf{s}_i)$. We define the $Q$-value of policy $\pi$ as the expected future success, measured by the final answer $a_L$ correctness, i.e.,

$$Q^\pi(\mathbf{s}_i, a_i) = Q^\pi\big((T, q, a_1, \ldots, a_{i-1}), a_i\big) = \mathbb{E}_{a_{i+1}, \ldots, a_L \sim \pi(\cdot \mid \mathbf{s}_i)}\left[\mathbb{1}_{a_L \text{ is correct}}\right]. \quad (4)$$

The value of policy $\pi$ at state $\mathbf{s}_i$ is defined as the expectation of $Q$-values under $\pi$'s next action distribution: $V^\pi(\mathbf{s}_i) = \mathbb{E}_{a_i \sim \pi(\cdot \mid \mathbf{s}_i)}[Q^\pi(\mathbf{s}_i, a_i)]$. We now analyze the policy improvement afforded by TATTOO's table-aware reward $r_i$ supervision under one step of a natural policy gradient updating.

---

**Theorem 4.1** (**Policy Improvement (Lower Bound)**). *Given the current policy $\pi$, after one natural policy gradient update step guided by the PRM reward $r_i$ defined in Eq.2, we obtain the revised policy $\pi'(a_i \mid \mathbf{s}_i) \propto \exp(Q^\pi(\mathbf{s}_i, a_i) + r_i(\mathbf{s}_i, a_i))$. The resulting expected policy improvement over the state distribution $\rho$ then satisfies:*

$$\mathbb{E}_{\mathbf{s}_i \sim \rho}\left[V^{\pi'}(\mathbf{s}_i) - V^\pi(\mathbf{s}_i)\right] \gtrsim \underbrace{\mathbb{E}_{\mathbf{s}_i \sim \rho}\text{Var}_{a_i \sim \pi(\cdot \mid \mathbf{s}_i)}\left[r_{i,tab}(\mathbf{s}_i, a_i)\right]}_{\text{distinguishability from table reward } r_{i,tab}} + \underbrace{\mathbb{E}_{\mathbf{s}_i \sim \rho}\text{Var}_{a_i \sim \pi(\cdot \mid \mathbf{s}_i)}\left[r_{i,rea}(\mathbf{s}_i, a_i)\right]}_{\text{distinguishability from inner-reasoning reward } r_{i,rea}}$$

$$+ \underbrace{\mathbb{E}_{\mathbf{s}_i \sim \rho}\mathbb{E}_{a_i \sim \pi(\cdot \mid \mathbf{s}_i)}\left[r_{i,tab}(\mathbf{s}_i, a_i)A^\pi(\mathbf{s}_i, a_i)\right]}_{\text{alignment between } r_{i,tab} \text{ and } A^\pi} + \underbrace{\mathbb{E}_{\mathbf{s}_i \sim \rho}\mathbb{E}_{a_i \sim \pi(\cdot \mid \mathbf{s}_i)}\left[r_{i,rea}(\mathbf{s}_i, a_i)A^\pi(\mathbf{s}_i, a_i)\right]}_{\text{alignment between } r_{i,rea} \text{ and } A^\pi},$$

$$(5)$$

*where $A^\pi(\mathbf{s}_i, a_i) = Q^\pi(\mathbf{s}_i, a_i) - V^\pi(\mathbf{s}_i)$ denotes the advantage of policy $\pi$.*

---

Theorem 4.1 (proof in Appendix E) explains that our decomposable reward design $r_i$ enables each component to additively contribute to policy improvement, provided that the reward components are each individually aligned with the policy advantage function. In this way, the table-aware rewards provided by TATTOO help ensure targeted supervision on both inner reasoning and table-involved operations generated by LRMs. Below, we further empirically evaluate the effectiveness of TATTOO across various downstream tabular reasoning tasks.

## 5 EMPIRICAL EVALUATIONS

**Baselines and Models.** We compare TATTOO against various types of step-level verification methods, including advanced ORMs/PRMs, majority voting (Liu et al., 2025), and LLM-as-a-judge (Zheng et al., 2023). The setups for these baselines are aligned with our preliminary analyses in Section 3. For PRM approaches, we include both discriminative (Qwen-PRM series (Zhang et al., 2025b), Math-Shepherd-PRM (Wang et al., 2024b), and Skywork-PRM (He et al., 2024a)) and generative (ThinkPRM (Khalifa et al., 2025) and GenPRM (Zhao et al., 2025)). Regarding the policy reasoning models, we evaluate our proposed method on DeepSeek-R1-Distill-Qwen-14B (Guo et al., 2025). Further details on the baselines and policy models setups are provided in Appendix F.1.

Table 2: Main results of TATTOO on 5 different tabular reasoning tasks. We report the best-of-N (with $N = \{4, 8, 16, 32\}$) performance using DeepSeek-R1-Distill-Qwen-14B as the policy model and compare against various step verifiers. The best and second-best results are highlighted. TATTOO consistently achieves state-of-the-art TTS performance with significantly fewer parameters.

| Verifer (Best-of-N) | Params | TB-NR | | | | TB-FC | | | | TB-DA | | | | WTQ | | | | MMQA | | | |
|---|---|---|---|---|---|---|---|---|---|---|---|---|---|---|---|---|---|---|---|---|---|
| | | 4 | 8 | 16 | 32 | 4 | 8 | 16 | 32 | 4 | 8 | 16 | 32 | 4 | 8 | 16 | 32 | 4 | 8 | 16 | 32 |
| Majority Vote | - | 65.5 | 65.9 | 66.8 | 66.5 | 76.2 | 77.3 | 77.3 | 77.4 | 23.5 | 24.5 | 26.0 | 26.1 | 64.7 | 65.3 | 67.3 | 67.0 | 18.4 | 19.4 | 20.4 | 20.1 |
| LLM-as-a-judge | - | 66.7 | 66.9 | 67.1 | 66.9 | 77.2 | 78.3 | 78.4 | 78.6 | 23.5 | 27.4 | 28.0 | 28.4 | 65.2 | 66.4 | 68.1 | 68.1 | 19.6 | 21.3 | 22.5 | 22.7 |
| Skywork-PRM-7B | 7B | 66.1 | 69.5 | 70.3 | 70.1 | 76.8 | 78.4 | 78.6 | 78.3 | 24.1 | 27.5 | 28.9 | 29.1 | 65.9 | 67.5 | 68.4 | 68.6 | 21.4 | 24.6 | 25.1 | 25.3 |
| Math-Shepherd-PRM-7B | 7B | 67.2 | 70.6 | 71.5 | 71.8 | 76.2 | 76.9 | 76.8 | 77.1 | 22.7 | 24.8 | 26.4 | 25.9 | 66.8 | 68.7 | 69.6 | 69.3 | 22.0 | 25.2 | 25.9 | 26.1 |
| Qwen2.5-Math-PRM-7B | 7B | 66.9 | 70.1 | 71.7 | 72.5 | 75.4 | 77.2 | 77.9 | 77.4 | 23.2 | 25.4 | 26.3 | 26.6 | 65.2 | 68.5 | 69.6 | 69.7 | 23.5 | 25.2 | 27.1 | 27.3 |
| ThinkPRM | 14B | 69.2 | 70.7 | 73.5 | 73.8 | 75.8 | 75.4 | 76.3 | 76.9 | 21.6 | 22.7 | 23.1 | 22.8 | 64.3 | 66.1 | 65.7 | 65.9 | 22.4 | 22.7 | 23.6 | 23.0 |
| GenPRM | 32B | 71.5 | 73.5 | 73.7 | 74.2 | 76.3 | 78.5 | 79.2 | 79.4 | 25.3 | 27.9 | 30.2 | 30.7 | 69.8 | 72.5 | 73.3 | 73.1 | 23.8 | 25.4 | 26.2 | 26.4 |
| Qwen2.5-Math-PRM-72B | 72B | 70.4 | 73.8 | 74.9 | 75.3 | 77.8 | 79.2 | 79.8 | 79.8 | 25.5 | 31.5 | 32.0 | 32.4 | 69.2 | 71.8 | 73.0 | 72.6 | 24.4 | 26.8 | 28.7 | 28.6 |
| TATTOO | 8B | 71.2 | 74.2 | 76.4 | 78.1 | 77.4 | 79.6 | 81.2 | 82.0 | 27.7 | 31.9 | 33.6 | 34.3 | 69.8 | 72.3 | 73.5 | 74.9 | 25.1 | 27.2 | 29.1 | 30.5 |

**Datasets.** We evaluate on four representative and challenging benchmarks spanning diverse tabular reasoning tasks, including (i) TableBench (TB) (Wu et al., 2024), a complex tabular reasoning benchmark with 886 questions covering tasks of numerical reasoning (NR), fact checking (FC), and data analysis (DA). (ii) WTQ (Pasupat and Liang, 2015b), a benchmark for complex question answering over Wikipedia tables. (iii) MMQA (Wu et al., 2025a), a multi-table understanding benchmark covering table retrieval, multi-hop & multi-table QA and text-to-SQL generation. We leave the additional dataset descriptions in Appendix F.2.

**Implementation Details.** We train TATTOO on the off-the-shelf Qwen-3-8B model (Yang et al., 2025a) using our 60k curated training instances (Section 4.2). All training and inference experiments are conducted on 8×A100-80G GPUs. To evaluate TATTOO under different TTS strategies, we adopt three representative methods, including Best-of-N (Brown et al., 2024), Beam Search (Snell et al., 2024), and Diverse Verifier Tree Search (DVTS) (Beeching et al., 2024). Additional implementation details on training setup and configurations of TATTOO are provided in Appendix F.3.

## 5.1 MAIN RESULTS

Table 2 reports the Best-of-N performance of incorporating TATTOO on the DeepSeek-R1-Distill-Qwen-14B model across five tabular reasoning tasks. Notably, TATTOO consistently outperforms strong baselines such as GenPRM-32B and Qwen2.5-Math-PRM-72B despite using only 8B parameters. On TB-DA, TATTOO achieves the largest accruacy performance across each level of N, rising from 27.7% at N=4 to 34.3% at N=32. Moreover, while existing PRMs often suffer from performance bottlenecks beyond a certain response threshold (as observed in Section 3), TATTOO continues to scale effectively, yielding consistent gains as the response group size increases. For example, on TB-NR, Qwen2.5-Math-PRM-72B saturates after N=16 (74.9% → 75.3%), whereas TATTOO continues to improve from 74.2% at N=8 to 78.1% at N=32. These results demonstrate that TATTOO provides stronger reward supervision on LRMs' reasoning trajectories, therefore yielding better performance improvement compared with other step-verification baselines.

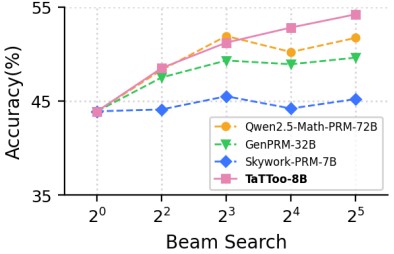
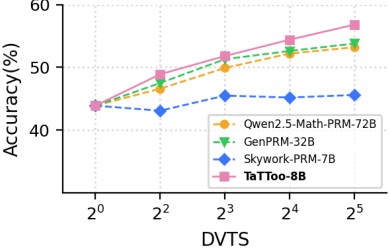

Figure 5: Performance of TATTOO on two additional TTS strategies, Beam Search and Diverse Verifier Tree Search (DVTS). We report the average accuracy across all 5 tabular reasoning tasks.

**Generalizability on Other TTS Strategies.** Beyond best-of-$N$, we also evaluate TATTOO under two additional TTS strategies (Beam Search and DVTS) and compare with the strongest PRM baselines. Figure 5 reports the average performance across the five tabular reasoning tasks. Under each TTS

Table 3: In-depth analysis of TATTOO on three table datasets. We evaluate the contributions of SFT and RL training stages, and assess the impact of reward shaping components during RL optimization.

| Training Variants | TB-NR | | | | TB-FC | | | | TB-DA | | | |
|---|---|---|---|---|---|---|---|---|---|---|---|---|
| | 4 | 8 | 16 | 32 | 4 | 8 | 16 | 32 | 4 | 8 | 16 | 32 |
| **TATTOO** *(SFT only)* | 67.9 | 69.1 | 72.0 | 73.7 | 71.5 | 73.0 | 74.6 | 75.2 | 23.3 | 25.6 | 26.2 | 26.4 |
| **TATTOO** | **71.2** | **74.2** | **76.4** | **78.1** | **77.4** | **79.6** | **81.2** | **82.0** | **27.7** | **31.9** | **33.6** | **34.3** |
| *w/o tool-grounding* | 68.5 | 71.1 | 72.7 | 74.6 | 73.2 | 75.6 | 75.5 | 76.3 | 26.2 | 28.1 | 28.7 | 30.3 |
| *w/o confidence calibration* | 71.1 | 73.7 | 74.3 | 76.2 | 76.4 | 76.7 | 78.4 | 80.5 | 27.4 | 29.5 | 31.3 | 33.2 |
| *rule-based (GRPO)* | 67.0 | 68.4 | 70.4 | 73.1 | 71.6 | 74.0 | 74.9 | 75.8 | 25.5 | 27.4 | 28.0 | 28.6 |

strategy, TATTOO consistently yields steady improvements as the number of responses N increases, whereas other baseline PRMs often plateau. For example, in beam search, TATTOO improves from 45.0% to 54.8%, while GenPRM saturates around 51% and Skywork-PRM remains below 46%. These results highlight the strong generalizability of TATTOO across diverse TTS strategies.

## 5.2 IN-DEPTH ANALYSES ON TATTOO

**Mastery of RL with Bootstrapping from SFT.** To examine the respective roles of SFT and RL in TATTOO's dual-stage training paradigm, we compare against a variant TATTOO (SFT only), which is trained solely on the first SFT stage. As shown in Table 3, under the Best-of-N evaluation, the second-stage RL policy optimization consistently improves performance over the SFT-only initialization. Specifically, we observe that the average accuracy across all three tasks improves from 72.3% (SFT only) to 78.5% after RL training, yielding a total gain of 10.2%. This demonstrates that bootstrapping from SFT provides a solid initialization, while RL optimization further enhances our PRM's reasoning and tool-use effectiveness during the verification process.

**Reward Shaping during RL Training.** Next, we analyze the effectiveness of each supervised component in our per-step reward signal $s_i$ design (Eq. 3), with the ablation results reported in Table 3. Removing the tool-grounding term yields the largest drop (e.g., ↓4.0% on TB-DA at N=32), highlighting its critical role in encouraging effective tool use during RL training. In addition, excluding confidence calibration reduces performance by 1.6% on average, showing its complementary effect in stabilizing reward signals. We also compare TATTOO with the original rule-based group-relative reward from GRPO, which yields only marginal improvement over SFT. Finally, Figure 6 visualizes the training dynamics of TATTOO and other variants during RL optimization.

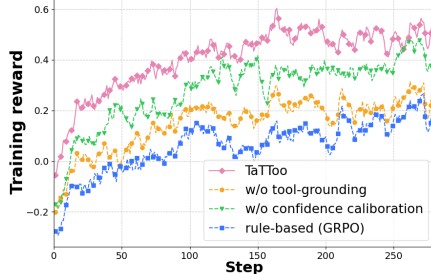

Figure 6: Training dynamics of TATTOO and ablated variants. We report the training reward across 280 training steps.

**Additional Experiments.** Additional experiments, including ablations on the training coefficients and case studies on TATTOO's effective tool usage, are provided in Appendix G.

## 6 RELATED WORKS

**LLMs on Tabular Reasoning.** Reasoning over tables poses a unique challenge for LLMs, requiring them to bridge natural language understanding with structured reasoning over rows, columns, and cell values (Jin et al., 2022; Zhang et al., 2025a). Recent works (Tang et al., 2020; Iida et al., 2021; Deng et al., 2022; Zou et al., 2025a) have investigated tabular reasoning on several downstream tasks, including table QA (Chen et al., 2020), table fact verification (Chen et al., 2019; Parikh et al., 2020), text-to-SQL (Mohammadjafari et al., 2024), etc. Early-stage tabular reasoning methods, such as TAPAS (Herzig et al., 2020) and TaBERT (Yin et al., 2020), encode table data into transformer-based encoder representations. Later methods leverage the capabilities of LLMs to apply either prompt engineering (Sui et al., 2023; Wang et al., 2024c) or supervised fine-tuning techniques (Su et al., 2024; Zhang et al., 2023) for enhanced tabular reasoning. More recent works, including the Table-R1

series (Wu et al., 2025b; Yang et al., 2025b; Jin et al., 2025) and Reasoning-Table (Lei et al., 2025), leverage RL to acquire higher-quality reasoning paths during reasoning over tables.

**Table Question Answering.** The evolution of Table Question Answering (Table QA) research (Jin et al., 2022) has been propelled by the creation of sophisticated evaluation resources that facilitate semantic parsing capabilities (Yang et al., 2020; Li et al., 2023b; 2024). Foundational works, including WTQ (Pasupat and Liang, 2015a) and TabFact (Chen et al., 2019), established initial evaluation paradigms through Wikipedia-derived HTML table QA pairs. Structured supervision has also been explored in alternative benchmarks such as WikiSQL (Zhong et al., 2017) and Spider (Yu et al., 2018), where logical expressions serve as explicit annotations to encourage systematic reasoning. More recent studies such as MultiTableQA (Wu et al., 2024), MT-RAIG (Seo et al., 2025), and MMQA (Wu et al., 2025a) has shifted towards multi-hop reasoning.

**PRMs for Test-time Scaling.** Process Reward Models (PRMs) (Lightman et al., 2024; Uesato et al., 2022; Zhang et al., 2024) deliver fine-grained, step-level feedback to guide model reasoning, assigning intermediate rewards to individual reasoning steps rather than only judging final answers (Guan et al., 2025; Chen et al., 2025). Prominent PRMs, including Math-Shepherd (Wang et al., 2024b), Skywork-PRM (He et al., 2024a), and the Qwen2.5-Math-PRM family (Zhang et al., 2025b), are trained using a mix of human annotations and synthesized supervision to score model-generated solution steps across domains such as math (Maxwell-Jia, 2024; Wang et al., 2026), scientific reasoning (Rein et al., 2023), and programming (He et al., 2024b); more recently, Think-PRM proposes a generative verifier to produce long-chain CoT evaluations (Khalifa et al., 2025). PRMs have been incorporated into training-time optimization as reward signals via step-verified online RL and verifier-guided self-training (Li and Li, 2024; Guan et al., 2025; Cui et al., 2025), and into inference-time scaling by coupling step-level scoring with search/decoding strategies (Zhao et al., 2025; Khalifa et al., 2025), including beam search, reward-guided tree search, and Best-of-N sampling. We leave additional related works on Process Reward Models and Tool Integration with RL in Appendix B.

## 7 CONCLUSION

We introduced TATTOO, a novel tool-augmented thinking PRM tailored for tabular reasoning. By diagnosing why existing verifiers fail on table retrieval and schema interaction, we built a scalable pipeline with expert rationales, table prefixes, and tool-augmented verification, and trained our model via SFT followed by RL with reward shaping. TATTOO achieves comparable performance across five table benchmarks, surpassing strong PRMs with up to 9× parameter efficiency and generalizing across multiple TTS strategies. Our results underscore the importance of table-grounded reward supervision and point toward future directions in reward modeling for structured reasoning tasks.

## ACKNOWLEDGMENT

This work is supported by National Science Foundation under Award No. IIS-2117902. The views and conclusions are those of the authors and should not be interpreted as representing the official policies of the funding agencies or the government.

## ETHICS STATEMENT

This work does not involve human subjects, sensitive personal information, or proprietary datasets. All datasets used in our experiments are publicly available table reasoning benchmarks, such as TabFact, FeTaQA, and WikiTableQuestions. We provide detailed descriptions of data processing steps in Section 5 and the Appendix D. The goal of our method is to improve process reward modeling for table reasoning, which we believe contributes positively to advancing trustworthy and interpretable reasoning with structured data. Nevertheless, we acknowledge that stronger table reasoning capabilities could be misused for generating misleading or biased tabular summaries if applied irresponsibly. We encourage responsible use and adherence to the ICLR Code of Ethics.

## REPRODUCIBILITY STATEMENT

We have made careful efforts to ensure reproducibility. The main text describes our model architecture, training procedure, and evaluation protocols in detail (Sections 4,5). Additional hyperparameters, implementation details, and ablation study configurations are included in the Appendix F. All datasets are publicly available, and preprocessing steps are documented in Section 5. Theoretical analyses, including formal definitions of preference-based reward modeling and proofs of consistency guarantees, are provided in Appendix E.

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

# TABLE OF CONTENTS

# Appendix

## A    THE USE OF LLMS

LLMs were used in this work in two main capacities. First, they served as the *base models* whose outputs were ranked and evaluated by our proposed TablePRM framework. We tested across a variety of publicly available pre-trained models, including Qwen and LLaMA families, to ensure robustness and generality, as described in Section 5 and Appendix F. Second, during the construction of preference pairs for supervision, we used LLM-generated responses as candidates, which were then compared and ranked according to factual consistency with gold tables. In addition, LLMs were used in a limited capacity for writing assistance, specifically to improve phrasing and readability of the manuscript. They did not contribute to research design, methodological innovations, or experimental results; all scientific contributions are the responsibility of the authors.

## B    ADDITIONAL RELATED WORK

**Discriminative vs. Generative PRM.**    In general, PRMs can be categorized as discriminative and generative evaluators (Zhong et al., 2025). A **discriminative PRM** treats verification as classification, directly predicting the correctness of each reasoning step with a scalar score. It is typically trained on step-level labels using cross-entropy loss, making it heavily reliant on step-level reward annotations. A **generative PRM** instead frames verification as conditional generation. It is trained with the standard language modeling objective to first generate rationales and then verify each step's correctness via a judgment token (e.g., [correct, incorrect]).

## C    DETAILED ERROR ANALYSIS

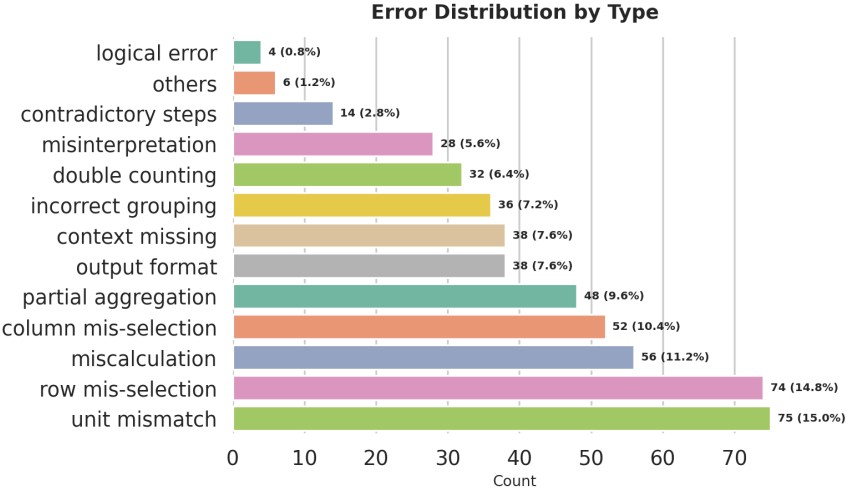

Figure 7: Error distribution over 500 incorrect LRM responses after Best-of-N. The errors are grouped into 13 predefined types, with the majority arising from table retrieval and schema interaction.

In Section 3, we perform a fine-grained error analysis on 500 erroneous responses sampled after Best-of-$N$ selection with Qwen2.5-Math-PRM-72B, to better understand the limitations of LRMs and PRMs. Each response is inspected and categorized by human experts into 13 predefined error types, covering both reasoning and table-specific mistakes. Figure 7 illustrates the overall error distribution.

**Error Type Distribution.**    The most frequent errors are *unit mismatch* (15.0%), *row mis-selection* (14.8%), and *miscalculation* (11.2%). Other common issues include *column mis-selection* (10.4%), *partial aggregation* (9.6%), and missing or incomplete *context* (7.6%). Less frequent but still notable categories include *output format errors*, *incorrect grouping*, *double counting*, *misinterpretation*, and

*contradictory steps*. A small portion of errors is grouped under *others* and *logical errors*. This diverse distribution highlights that model failures are not restricted to arithmetic slips but extend to schema understanding and structural reasoning.

**Mapping to Reasoning-Step Categories.**   To reveal deeper patterns, we align the 13 error types with four reasoning-step categories reflecting the typical flow of LRMs:

- **Table Retrieval Step**: Includes row/column mis-selection, unit mismatch, and partial aggregation. These account for 47.7% of total errors, indicating difficulty in locating and extracting the correct table region.
- **Schema Interaction Step**: Covers miscalculation, grouping mistakes, double counting, and misinterpretation of table semantics. This represents 34.3% of errors, reflecting challenges in reasoning over structured contents once retrieved.
- **Inner-Thinking Step**: Logical errors or contradictory reasoning steps independent of table contents. These contribute 12.0% of total errors, suggesting LRMs remain relatively competent in pure logical chains compared to table-centric operations.
- **Others**: Errors arising from context omission or improper output formatting.

**Key Findings.**   The analysis confirms that most model weaknesses lie in table-related operations, including table retrieval and schema interaction, rather than general logical reasoning. PRMs, when supervising such steps, face greater challenges since they must not only validate the correctness of reasoning but also verify alignment between the retrieved sub-table and the query.

## D  TATTOO DATA CURATION PIPLINE

We design a large-scale data curation pipeline that simulates real-world scenarios of PRM tool use and step verification at scale. As illustrated in Figure 1, there are three main stages:

**Reasoning Trajectory Generation.** We begin by collecting trajectory responses from expert LRMs (e.g., DeepSeek-R1 (Guo et al., 2025) and Claude-Opus-4.1 (Anthropic, 2025)) on table-based questions drawn from diverse benchmarks, including TableInstruct (Wu et al., 2024), HybridQA (Chen et al., 2020), ToTTo (Parikh et al., 2020), and WikiTQ (Pasupat and Liang, 2015b).

We generate multiple model responses per query, capturing both correct and incorrect reasoning patterns. We then adopt a dual-verification procedure (Feng et al., 2025a), where both human annotators and expert LLMs are employed to examine and filter out low-quality or incomplete CoT data. Through this, we receive a high-quality set of LRMs' output responses $\mathcal{T}_{\text{pool}}$ for subsequent data labeling.

**Verification Synthesis & Reward Assignment.** Our next step is to provide step-level verification rationales and assign PRM step-reward labels for each candidate response in $\mathcal{T}_{\text{pool}}$. To this end, we first identify the table retrieval and schema interaction steps within each response in $\mathcal{T}_{\text{pool}}$:

*Table retrieval steps* - We first extract the retrieved sub-table from each step. Then we apply LLM-as-a-judge to evaluate whether retrieved contents are accurate and provide complete rationales for the judgment. We assign step-level table reward $r_{i,\text{tab}} \in \{-1, 1\}$ (in Eq. 1) based on the correctness of the retrieval, while setting $r_{i,\text{rea}}$ to 0. This reward supervision explicitly trains PRMs to recognize if the retrieved sub-table aligns with the input query, addressing the limitation shown in *Takeaway 1*.

*Schema interaction steps* - We collect the sub-table retrieved from the preceding table retrieval step and use it as a table prefix. If the retrieval is incorrect, we manually replace it with the correct sub-table corresponding to the query. We then prepend this table prefix to the verification rationale generated by LLM-as-a-judge. Finally, we assign the PRM's step-level table reward $r_{i,\text{tab}} \in \{-1, 1\}$ based on the correctness of the schema interaction, and $r_{i,\text{rea}}$ to 0. By explicitly attaching the retrieved sub-table to each schema interaction step, we mitigate the dependencies issue noted in *Takeaway 2*.

*Other steps without table operations involved* - We directly query an expert LLM (DeepSeek-R1) to generate verification rationales. We assign the PRM's step-level reasoning reward $r_{i,\text{rea}} \in \{-1, 1\}$ based on the correctness of the reasoning, while setting the table reward $r_{i,\text{tab}}$ to 0.

**Tool Use Synthesis.** To help PRMs learn to leverage tools for more accurate verification, we augment the collected verification rationales by incorporating tool invocation, execution, and feedback into the verification steps. Specifically, whenever the model's inner reasoning involves a calculation or table lookup operation, we replace it with the corresponding tool call and its execution result. We primarily employ two types of tools:

*Computation tools* - Applying Python or SQL code snippets for arithmetic or aggregation operations. E.g., if a step verifies the sum of a table column, we replace the model's manual calculation with a code snippet that executes the summation and returns the result.

*Table lookup tools* - Locating and extracting specific rows, columns, or cells from the table. E.g., if a step requires referencing a sub-table cell value during the verification, we replace the model's self-extraction with an explicit lookup tool call that retrieves the corresponding entry.

By integrating verification processes with code snippets and real-time interpreter feedback, we construct roughly 60k data for TATTOO's verification reasoning and tool usage.

# E   PROOF OF THEOREM 4.1

**Notational conventions.**   We use $\mathbf{s}_i$ for a state, $a_i$ for an action, $\pi$ for the current policy, and $\pi'$ for the updated policy. The advantage is defined as

$$A^\pi(\mathbf{s}_i, a_i) = Q^\pi(\mathbf{s}_i, a_i) - V^\pi(\mathbf{s}_i). \tag{6}$$

The PRM signal at a step is the overall process reward, defined in Eq.2. For a fixed $\mathbf{s}_i$, we write $\mathbb{E}_\pi[\cdot] \equiv \mathbb{E}_{a_i \sim \pi(\cdot|\mathbf{s}_i)}[\cdot]$, $\mathrm{Var}_\pi[\cdot] \equiv \mathrm{Var}_{a_i \sim \pi(\cdot|\mathbf{s}_i)}[\cdot]$, and $\mathrm{Cov}_\pi(r_{i,\mathrm{rea}}(\mathbf{s}_i, a_i), r_{i,\mathrm{rea}}(\mathbf{s}_i, a_i)) \equiv \mathrm{Cov}_{a_i \sim \pi(\cdot|\mathbf{s}_i)}(r_{i,\mathrm{rea}}(\mathbf{s}_i, a_i), r_{i,\mathrm{rea}}(\mathbf{s}_i, a_i))$ Expectations over states use the subscript explicitly, e.g., $\mathbb{E}_{\mathbf{s}_i \sim \rho}[\cdot]$. We use $d_\rho^{\pi'}$ for the state distribution induced by the policy $\pi'$ starting from the initial distribution $\rho$. Finally, $X \gtrsim Y$ means there exists a universal constant $c > 0$, independent of $(\pi, \pi', \mathbf{s}_i)$, such that $X \geq cY$.

We start the proof by introducing two standard lemmas that will be used repeatedly; both are well-known results in the RL literature, and we omit their proofs here for brevity.

**Lemma E.1** (**Performance Difference Lemma (PDL)**). *For any pair of policies $\pi$ and $\pi'$ defined over the same Markov decision process with initial state distribution $\rho$, the following identity holds:*

$$\mathbb{E}_{\mathbf{s}_i \sim \rho}\Big[V^{\pi'}(\mathbf{s}_i) - V^\pi(\mathbf{s}_i)\Big] \;=\; \mathbb{E}_{\mathbf{s}_i \sim d_\rho^{\pi'}} \mathbb{E}_{a_i \sim \pi'(\cdot|\mathbf{s}_i)}\big[A^\pi(\mathbf{s}_i, a_i)\big].$$

See proof of Lemma 6.1 in (Kakade and Langford, 2002).

**Lemma E.2** (**Natural policy gradient (NPG) update form**). *Fix a step size $\gamma > 0$. If the NPG update is guided by the signal $A^\pi(\mathbf{s}_i, a_i) + r_i(\mathbf{s}_i, a_i)$, then*

$$\pi'(a_i \mid \mathbf{s}_i) \propto \pi(a_i \mid \mathbf{s}_i) \exp\Big(\gamma\big(A^\pi(\mathbf{s}_i, a_i) + r_i(\mathbf{s}_i, a_i)\big)\Big),$$

$$Z^\pi(\mathbf{s}_i) \triangleq \sum_{a_i} \pi(a_i \mid \mathbf{s}_i)\Big[\exp\Big(\gamma\big(A^\pi(\mathbf{s}_i, a_i) + r_i(\mathbf{s}_i, a_i)\big)\Big)\Big], \tag{7}$$

$$\text{so that}\quad \frac{\pi'(a_i \mid \mathbf{s}_i)}{\pi(a_i \mid \mathbf{s}_i)} = \frac{\exp\Big(\gamma\big(A^\pi(\mathbf{s}_i, a_i) + r_i(\mathbf{s}_i, a_i)\big)\Big)}{Z^\pi(\mathbf{s}_i)}.$$

See proof of Lemma F.2 in (Setlur et al., 2024). Next, we restate Theorem 4.1 in the following proposition.

**Proposition E.3** (**Full-strength policy improvement lower bound**). *Let $\pi'$ be the NPG update in Lemma E.2. We can have:*

$$\mathbb{E}_{\mathbf{s}_i \sim \rho}\Big[V^{\pi'}(\mathbf{s}_i) - V^\pi(\mathbf{s}_i)\Big] \;\gtrsim\; \mathbb{E}_{\mathbf{s}_i \sim \rho}\Big[\; \underbrace{\mathrm{Var}_\pi\big[r_{i,\mathrm{rea}}(\mathbf{s}_i, a_i)\big]}_{\textit{distinguishability (reasoning reward)}} \;+\; \underbrace{\mathrm{Var}_\pi\big[r_{i,\mathrm{tab}}(\mathbf{s}_i, a_i)\big]}_{\textit{distinguishability (table reward)}}$$

$$+2\underbrace{\mathrm{Cov}_\pi\big(r_{i,rea}(\mathbf{s}_i, a_i),\, r_{i,tab}(\mathbf{s}_i, a_i)\big)}_{\textit{alignment between } r_{i,rea} \textit{ and } r_{i,tab}} + \underbrace{\mathbb{E}_\pi\big[r_{i,tab}(\mathbf{s}_i, a_i)\, A^\pi(\mathbf{s}_i, a_i)\big]}_{\textit{alignment of } r_{i,tab} \textit{ with } A^\pi} + \underbrace{\mathbb{E}_\pi\big[r_{i,rea}(\mathbf{s}_i, a_i)\, A^\pi(\mathbf{s}_i, a_i)\big]}_{\textit{alignment of } r_{i,rea} \textit{ with } A^\pi}\Big]. \tag{8}$$

*Proof of Proposition E.3.* We now combine the performance difference lemma with the NPG update to derive a variance–alignment lower bound, while first retaining the covariance term between the reward components. By Lemma E.1, we have

$$\mathbb{E}_{\mathbf{s}_i \sim \rho}\big[V^{\pi'}(\mathbf{s}_i) - V^\pi(\mathbf{s}_i)\big] \;=\; \mathbb{E}_{\mathbf{s}_i \sim d_\rho^{\pi'}} \mathbb{E}_{a_i \sim \pi'(\cdot|\mathbf{s}_i)}\big[A^\pi(\mathbf{s}_i, a_i)\big]. \tag{9}$$

**Exponential tilting and a log-partition bound.** Let us define the log-partition at state $\mathbf{s}_i$ by

$$\log Z^\pi(\mathbf{s}_i) \;=\; \log \mathbb{E}_{a_i \sim \pi(\cdot|\mathbf{s}_i)} \exp\Big(\gamma\big(A^\pi(\mathbf{s}_i, a_i) + r_i(\mathbf{s}_i, a_i)\big)\Big).$$

From Lemma E.2, we have

$$A^\pi(\mathbf{s}_i, a_i) = \frac{1}{\gamma} \log \frac{\pi'(a_i \mid \mathbf{s}_i)}{\pi(a_i \mid \mathbf{s}_i)} - r_i(\mathbf{s}_i, a_i) + \frac{1}{\gamma} \log Z^\pi(\mathbf{s}_i).$$

Averaging over $a_i \sim \pi'(\cdot \mid \mathbf{s}_i)$, using $\mathbb{E}_{\pi'}[\log \frac{\pi'}{\pi}] \geq 0$, Jensen's inequality on $\log Z^\pi(\mathbf{s}_i)$ and $\mathbb{E}_\pi[A^\pi(\mathbf{s}_i, a_i)] = 0$ gives

$$\mathbb{E}_{a_i \sim \pi'(\cdot | \mathbf{s}_i)}[A^\pi(\mathbf{s}_i, a_i)] \geq -\mathbb{E}_{a_i \sim \pi'(\cdot | \mathbf{s}_i)}[r_i(\mathbf{s}_i, a_i)] + \mathbb{E}_{a_i \sim \pi(\cdot | \mathbf{s}_i)}[r_i(\mathbf{s}_i, a_i)]. \tag{10}$$

Plugging this into equation 9 yields the basic inner-product lower bound

$$\mathbb{E}_{\mathbf{s}_i \sim \rho}\big[V^{\pi'}(\mathbf{s}_i) - V^\pi(\mathbf{s}_i)\big] \geq \mathbb{E}_{\mathbf{s}_i \sim d_\rho^{\pi'}}\langle \pi'(\cdot \mid \mathbf{s}_i) - \pi(\cdot \mid \mathbf{s}_i), r_i(\mathbf{s}_i, \cdot)\rangle. \tag{11}$$

Using first-order expansion of the exponential tilt implies

$$\langle \pi'(\cdot \mid \mathbf{s}_i) - \pi(\cdot \mid \mathbf{s}_i), r_i(\mathbf{s}_i, \cdot)\rangle \gtrsim \big(\mathrm{Var}_\pi[r_i(\mathbf{s}_i, a_i)] + \mathbb{E}_\pi[r_i(\mathbf{s}_i, a_i) A^\pi(\mathbf{s}_i, a_i)]\big), \tag{12}$$

Combining equation 11 and equation 12, and weakening $d_\rho^{\pi'}$ to $\rho$ (componentwise monotonicity) gives

$$\mathbb{E}_{\mathbf{s}_i \sim \rho}\big[V^{\pi'}(\mathbf{s}_i) - V^\pi(\mathbf{s}_i)\big] \gtrsim \mathbb{E}_{\mathbf{s}_i \sim \rho}\Big[\mathrm{Var}_\pi[r_i(\mathbf{s}_i, a_i)] + \mathbb{E}_\pi[r_i(\mathbf{s}_i, a_i) A^\pi(\mathbf{s}_i, a_i)]\Big]. \tag{13}$$

**Variance decomposition with covariance.** Next, using $r_i = r_{i,\mathrm{rea}} + r_{i,\mathrm{tab}}$, we have

$$\mathrm{Var}_\pi[r_i(\mathbf{s}_i, a_i)] = \mathrm{Var}_\pi[r_{i,\mathrm{rea}}(\mathbf{s}_i, a_i)] + \mathrm{Var}_\pi[r_{i,\mathrm{tab}}(\mathbf{s}_i, a_i)] + 2\,\mathrm{Cov}_\pi(r_{i,\mathrm{rea}}(\mathbf{s}_i, a_i), r_{i,\mathrm{tab}}(\mathbf{s}_i, a_i)). \tag{14}$$

Substituting into equation 13 complete our proof of Proposition E.3 (equation 8). $\qquad\square$

**Covariance elimination under our reward design.** By construction in our setup (see Section 4.2), for each state–action pair $(\mathbf{s}_i, a_i)$, the two components of the PRM signal, i.e., table reward and reasoning reward, are *mutually exclusive*. Formally, we have

$$r_{i,\mathrm{tab}}(\mathbf{s}_i, a_i) \in \{-1, 0, 1\}, \quad r_{i,\mathrm{rea}}(\mathbf{s}_i, a_i) \in \{-1, 0, 1\}, \quad \text{and} \quad r_{i,\mathrm{tab}}(\mathbf{s}_i, a_i)\, r_{i,\mathrm{rea}}(\mathbf{s}_i, a_i) = 0.$$

Policy-gradient updates are invariant to adding any per-state baseline, so we may center each component without loss, i.e.,

$$\tilde{r}_{i,\mathrm{rea}}(\mathbf{s}_i, a_i) = r_{i,\mathrm{rea}}(\mathbf{s}_i, a_i) - \mathbb{E}_\pi[r_{i,\mathrm{rea}}(\mathbf{s}_i, a_i)], \qquad \tilde{r}_{i,\mathrm{tab}}(\mathbf{s}_i, a_i) = r_{i,\mathrm{tab}}(\mathbf{s}_i, a_i) - \mathbb{E}_\pi[r_{i,\mathrm{tab}}(\mathbf{s}_i, a_i)].$$

Mutual exclusivity yields $\mathbb{E}_\pi[\tilde{r}_{i,\mathrm{rea}}(\mathbf{s}_i, a_i)\, \tilde{r}_{i,\mathrm{tab}}(\mathbf{s}_i, a_i)] = 0$, hence $\mathrm{Cov}_\pi(\tilde{r}_{i,\mathrm{rea}}, \tilde{r}_{i,\mathrm{tab}}) = 0$ and

$$\mathrm{Var}_\pi[\tilde{r}_i(\mathbf{s}_i, a_i)] = \mathrm{Var}_\pi[\tilde{r}_{i,\mathrm{rea}}(\mathbf{s}_i, a_i)] + \mathrm{Var}_\pi[\tilde{r}_{i,\mathrm{tab}}(\mathbf{s}_i, a_i)], \quad \tilde{r}_i \triangleq \tilde{r}_{i,\mathrm{rea}} + \tilde{r}_{i,\mathrm{tab}}.$$

Plugging these centered quantities into the bounds of Proposition E.3 (which is NPG-invariant under per-state centering) gives exactly Theorem 4.1's inequality:

$$\begin{aligned}
\mathbb{E}_{\mathbf{s}_i \sim \rho}\Big[V^{\pi'}(\mathbf{s}_i) - V^\pi(\mathbf{s}_i)\Big] &\gtrsim \mathbb{E}_{\mathbf{s}_i \sim \rho}\Big[\mathrm{Var}_\pi[r_{i,\mathrm{rea}}(\mathbf{s}_i, a_i)] + \mathrm{Var}_\pi[r_{i,\mathrm{tab}}(\mathbf{s}_i, a_i)] \\
&\quad + \mathbb{E}_\pi[r_{i,\mathrm{tab}}(\mathbf{s}_i, a_i) A^\pi(\mathbf{s}_i, a_i)] + \mathbb{E}_\pi[r_{i,\mathrm{rea}}(\mathbf{s}_i, a_i) A^\pi(\mathbf{s}_i, a_i)]\Big],
\end{aligned} \tag{15}$$

which completes the proof of Theorem 4.1. $\qquad\square$

**Remarks.** (i) Proposition E.3 is strictly more general; Theorem 4.1 follows as a corollary under mutual exclusivity plus per-state centering (baseline invariance). (ii) Mutual exclusivity alone yields $\mathbb{E}_\pi[r_{i,\mathrm{rea}}\, r_{i,\mathrm{tab}}] = 0$, but per-state centering is what ensures $\mathrm{Cov}_\pi(r_{i,\mathrm{rea}}, r_{i,\mathrm{tab}}) = 0$. (iii) The alignment term necessarily uses the composite signal $r_i$ because the NPG step is guided by $A^\pi + r_i$.

# F    EXPERIMENTAL SETUPS

## F.1    POLICY MODEL CONFIGURATIONS

In our experiments, we adopt an LRM DeepSeek-R1-Distill-Qwen-14B (Guo et al., 2025) as the downstream policy model. During inference, we configure the model with a temperature of 0.7, a maximum generation length of 16,384 tokens, and top-$p$ sampling with $p = 0.95$. We evaluate the LRM on several inference-time scaling strategies:

**Best-of-N (BoN).**    The policy model generates $N$ candidate responses independently. A verifier (PRM) scores each response, and the final output is selected based on a voting or scoring method.

**Beam Search.**    Given beam width $N$ and branching factor $M$, the model generates $N$ initial steps. The verifier then selects the top $N/M$ continuations, and the model expands each with $M$ new candidates. This process repeats until termination, enabling guided exploration of high-quality reasoning paths.

**Diverse Verifier Tree Search (DVTS).**    DVTS is a variant of beam search where the search process is divided into multiple subtrees. Each subtree is explored independently using verifier-guided expansions, with candidates selected at every step based on PRM scores.

**Majority Voting.**    After generating multiple responses, the final answer is determined by simple majority over identical outputs, regardless of intermediate step scores. This method provides a baseline aggregation mechanism.

**LLM-as-a-Judge.**    Instead of relying solely on PRMs, a separate LLM is prompted to compare and evaluate candidate responses directly, selecting the most plausible or logically consistent output.

## F.2    EVALUATION DATASET DETAILS

**TableBench (Wu et al., 2024).**    TableBench is a comprehensive benchmark specifically designed to evaluate the reasoning abilities of LLMs over tabular data. It consists of 3,681 unique tables drawn from diverse domains such as finance, sports, politics, and science, with each table containing on average 16.7 rows and 6.7 columns. The dataset emphasizes numerical reasoning, with over $65\%$ of table cells containing numerical values. TableBench questions are organized into four major categories: fact-checking, numerical reasoning, data analysis, further divided into 18 subcategories, yielding a total of 886 carefully annotated samples. Each question typically requires 6.3 reasoning steps, making the dataset significantly more complex than prior TableQA corpora.

**WikiTableQuestions (WTQ) (Pasupat and Liang, 2015b).**    The WikiTableQuestions dataset introduces question answering over semi-structured HTML tables, aiming to test both compositional reasoning and domain generalization. It comprises 22,033 natural language questions paired with 2,108 Wikipedia tables, where the training and test tables are disjoint to ensure generalization to unseen schemas. The tables are semi-structured and heterogeneous, often containing multi-part cell values (e.g., "Beijing, China") that require normalization into multiple semantic types such as numbers or dates. Questions range from simple lookups to highly compositional queries involving comparison, aggregation, arithmetic, and superlatives. Each table contains at least 8 rows and 5 columns, and the question collection was conducted with quality control through multiple annotators.

**MMQA (Wu et al., 2025a)**    MMQA is a large-scale benchmark for evaluating LLMs on multi-table and multi-hop question answering. The benchmark includes a total of 3,312 relational tables across 138 domains, where each instance consists of two or three interlinked tables. The dataset features 5,000 multi-table samples, annotated with natural language questions, SQL queries, gold answers, and explicit primary/foreign key relations. To ensure annotation quality, foreign and primary keys were labeled by human experts with inter-annotator agreement exceeding $80\%$. MMQA questions span four main categories, including numerical, list, count, and select, with an average length of 77–85 tokens, reflecting their compositional complexity.

Table 4: Comparison of TATTOO with Output-Reward-Model (ORM) baselines.

| Method (Best-of-16) | TB-NR | TB-FC | TB-DA | WTQ | MMQA |
|---|---|---|---|---|---|
| Discriminative ORM | 66.4 | 72.0 | 26.8 | 68.1 | 25.3 |
| Generative ORM | 70.6 | 75.9 | 28.5 | 69.2 | 26.6 |
| **TaTToo** | **76.4** | **81.2** | **33.6** | **73.5** | **29.1** |

### F.3 TRAINING DETAILS

We train TATTOO using the off-the-shelf Qwen-3-8B model (Yang et al., 2025a) on our curated 60k dataset. For supervised fine-tuning, we adopt the LLaMA-Factory framework (Zheng et al., 2024). The training setup uses a learning rate of $1 \times 10^{-5}$, a weight decay of $1 \times 10^{-4}$, a maximum sequence length of 20,000, and is run for 3 epochs. For the RL training stage, we adopt the VeRL framework (Sheng et al., 2024) to further optimize the SFT checkpoint via policy optimization. The model is trained with a batch size of 32, generating 8 samples per question as the group size, and is run for 3 epochs. During inference, we use the OpenR framework (Wang et al., 2024a) to deploy our trained TATTOO-8B, which serves as a verifier to guide the downstream LRM under different test-time scaling strategies.

## G ADDITIONAL EXPERIMENTS

### G.1 COMPARISON WITH ORM BASELINES

To evaluate the efficacy of our approach, we compare TaTToo against two distinct Outcome Reward Model (ORM) baselines, both implemented on the same Qwen3-8B backbone. The comparison is reported in Table 4. We detail the training and implementation settings for these ORM baselines below. For the policy model and all additional configurations, we follow exactly the same setup as outlined in our main experiments. First, following (Mahan et al., 2024) and our established experimental settings, we employ a **Generative ORM** trained via a dual-stage process (SFT followed by RL) using our curated dataset. In this configuration, the model is optimized to generate a rationale chain before producing a final output-level reward token ("correct" or "incorrect") for each complete instance. Second, adhering to the methodologies of (Hosseini et al., 2024; Cobbe et al., 2021), we implement a **Discriminative ORM** by training a classification head directly on the backbone model. This baseline is designed to classify whether a candidate answer's final solution is correct or incorrect without generating any intermediate rationales.

In Table 4, we show that TaTToo provides significantly stronger supervision than both ORM baselines. This performance advantage indicates that TaTToo's process-level supervision delivers denser and richer reward modeling signals, which in turn contribute positively to the performance of the downstream policy model. Furthermore, when comparing the two baselines, the Generative ORM consistently outperforms the Discriminative ORM. This gap suggests that the inclusion of generative rationales offers more informative supervision than binary correctness labels alone. As the Generative ORM effectively leverages the reasoning paths within our training data, this result further highlights the rich verification rationales and high-quality supervision signals provided by our curated dataset.

Table 5: Performance of Table-R1-Zero with and without TaTToo under the Best-of-$N$ setting. TaTToo consistently improves the RL-trained table reasoner across all tasks.

| Method | TB-NR | TB-FC | TB-DA | WTQ | MMQA |
|---|---|---|---|---|---|
| Table-R1-Zero | 34.8 | 61.6 | 16.4 | 77.3 | 24.2 |
| Table-R1-Zero + TaTToo (BoN-4) | 39.6 | 64.7 | 18.3 | 80.9 | 26.8 |
| Table-R1-Zero + TaTToo (BoN-8) | 45.1 | 69.0 | 20.1 | 82.6 | 28.4 |
| Table-R1-Zero + TaTToo (BoN-16) | 48.2 | 72.3 | 23.5 | 84.5 | 30.3 |

## G.2 TaTToo on RL-Trained Table-R1

To demonstrate that TaTToo is complementary to RL-based table-reasoning methods, we further evaluate its performance when paired with Table-R1-Zero (Yang et al., 2025b), a 7B table-specialist policy model trained with program-based reinforcement learning. Under the Best-of-$N$ test-time sampling strategy ($N \in \{4, 8, 16\}$), we apply TaTToo as the verifier following the same evaluation protocol described in Section 5 and compare Table-R1-Zero alone against Table-R1-Zero augmented with TaTToo across all five tasks. As shown in Table 5, incorporating TaTToo consistently improves accuracy and yields larger gains as $N$ increases, indicating that TaTToo reliably identifies higher-quality trajectories during test-time scaling. These results further confirm that TaTToo is architecture-agnostic and serves as a flexible verifier that enhances diverse table-reasoning systems, including RL-trained approaches such as Table-R1.

## G.3 Ablation Study on TaTToo

Table 6: Ablation on confidence calibration $\lambda_{\text{cal}}$.

| N=32 | TB-NR | TB-FC | TB-DA |
|------|-------|-------|-------|
| 0.3 | 76.8 | 80.9 | 33.1 |
| 0.5 | 77.3 | 81.3 | 33.6 |
| 0.8 | 78.1 | **82.0** | **34.3** |
| 1.0 | **78.5** | 81.4 | 33.8 |

Table 7: Ablation on tool-grounding $\lambda_{\text{tool}}$.

| N=32 | TB-NR | TB-FC | TB-DA |
|------|-------|-------|-------|
| 0.1 | 75.2 | 76.3 | 30.8 |
| 0.5 | 75.9 | 76.9 | 32.2 |
| 1.0 | **78.1** | **82.0** | 34.3 |
| 1.3 | 77.5 | 81.2 | **34.6** |

**Ablations on $\lambda_{\text{cal}}$ and $\lambda_{\text{tool}}$.** In Eq. 3, we use $\lambda_{\text{cal}}$ and $\lambda_{\text{tool}}$ as tunable coefficients to balance the contributions of the corresponding reward terms in GRPO. To examine their influence, we separately train our verifier model (initialized from the same SFT checkpoint) by varying $\lambda_{\text{cal}} \in \{0.3, 0.5, 0.8, 1.0\}$ and $\lambda_{\text{tool}} \in \{0.1, 0.5, 1.0, 1.5\}$ during RL, and then evaluate on TableBench with N = 32. As shown in Table 6 and 7, performance improves as $\lambda_{\text{cal}}$ increases, peaking at 0.8–1.0. For $\lambda_{\text{tool}}$, accuracy rises steadily and is strongest around 1.0–1.3. These results empirically confirm the effectiveness of confidence calibration and tool-grounding in enhancing TTS.

## G.4 Efficiency Analyses on TaTToo

To ensure computational efficiency, our design prioritizes concise data curation. As detailed in Section 4.2, we implement a progressive filtering strategy that yields a high-quality training set of approximately 60,000 examples. This dataset size is significantly more compact than standard Process Reward Model (PRM) corpora; for comparison, the baseline Qwen-PRM models utilized in our experiments required roughly 800,000 training samples (Lightman et al., 2023; Zhang et al., 2025b). Leveraging this compact dataset alongside a lightweight backbone allows the complete TaTToo training process to conclude in fewer than 8 GPU hours. Beyond efficiency, the resulting PRM demonstrates robust generalization capabilities. Once trained, TaTToo provides effective step-level supervision across a diverse spectrum of tabular reasoning tasks, including question answering, fact-checking, and data analysis, while still remaining compatible with various Test-Time Selection (TTS) strategies such as Best-of-N and Beam Search. This versatility underscores TaTToo's broad applicability across the tabular reasoning domain.

Table 8: Computational cost breakdown of TaTToo training stages.

| TaTToo Training Stage | Training Data Size | # of Training Steps | GPU Setup | Total Hours | Total Cost |
|------------------------|--------------------|--------------------|-----------|-------------|------------|
| SFT | 50K | ∼2400 | 8×A100 | 5.4 | $25.9 |
| RL | 10K | ∼280 | 8×A100 | 8.3 | $39.8 |

To quantify the accessibility of our method, we analyze the training overhead of TaTToo. We present a detailed cost breakdown for the 8B model, encompassing both the Supervised Fine-Tuning (SFT) and Reinforcement Learning (RL) stages. To provide a tangible economic metric, we evaluate the total GPU computational cost based on a standardized pricing rate of $4.8 per 8-GPU hours.

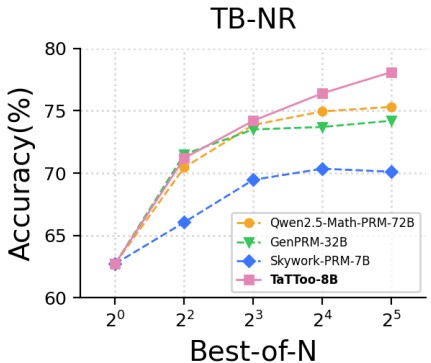

Figure 8: Performance of TᴀTTᴏᴏ and baseline PRMs on TB-NR under Best-of-N test time scaling. While baseline models plateau as $N$ increases, TᴀTTᴏᴏ continues to scale effectively, yielding consistent accuracy gains.

While the RL stage utilizes only approximately one-fifth of the data and one-ninth of the training steps relative to the SFT phase, it incurs a higher computational cost due to the overhead associated with rollout sampling and tool executions. However, the absolute duration of the RL stage remains limited. As demonstrated in Table 8, given the substantial performance improvements observed over the SFT-only baseline, this additional computational expenditure represents a favorable efficiency-effectiveness tradeoff. Overall, the complete TaTToo training pipeline requires approximately 14 GPU hours, translating to a cost of roughly $65. This resource requirement is considerably lower than that of training larger PRM baselines, such as those with 32B or 72B parameters.

### G.5 Performance Gain of TᴀTTᴏᴏ with Increasing Number of Responses

Figure 8 presents the best-of-$N$ performance on TB-NR. We observe that baseline PRMs such as Qwen2.5-Math-PRM-72B and GenPRM-32B quickly saturate beyond $N{=}16$, achieving only marginal improvements at larger $N$. Skywork-PRM-7B shows even weaker scalability, plateauing below 71%. In contrast, TᴀTTᴏᴏ continues to improve steadily as $N$ increases, reaching 78.3% at $N{=}32$, the highest among all models despite having significantly fewer parameters (8B). These results highlight the scalability advantage of TᴀTTᴏᴏ, demonstrating its ability to leverage larger response pools more effectively than existing PRMs.

## H  Case Study on TaTToo

As shown in Figure 9, we present a case study on TᴀTTᴏᴏ illustrating the difference between the verification processes at the two training stages on a specific instance in Figure 9 (Appendix H). When facing the same step (Step 3), the SFT-stage relies on inner text reasoning to verify the calculation, but introduces numerical errors that lead to incorrect justification of the step's correctness. In contrast, the RL-stage learns to leverage the computation tool with concise Python code, ensuring accurate calculations and thereby providing more reliable reward supervision on the policy model's responses. In addition, we randomly sample 500 trajectories from both stages of TᴀTTᴏᴏ on the same set of inputs and observe a 26.3% improvement in the tool-integration ratio after RL training, indicating our model learns to utilize tools better for step-level verification during RL rollouts.

## I    LIMITATIONS AND BROADER IMPACTS

While reinforcement learning with reward shaping enhances our PRM's ability to capture fine-grained tabular reasoning signals, it introduces more computational overhead. Compared to SFT-only training, the RL stage requires additional rollouts, reward evaluations, and optimization steps, which can increase training cost and resource demands. This overhead may hinder reproducibility and accessibility in low-resource environments, motivating future work on more efficient reward objectives and lightweight reward modeling strategies. In addition, our current framework is limited to text–table reasoning, and extending it to multimodal settings (e.g., integrating charts or image-based tables) remains an important direction for future work.

From a broader perspective, this work highlights the potential for process reward models to enhance structured reasoning in domains such as fact-checking, scientific analysis, and decision support. At the same time, reliance on automated verification carries risks: if tools or training data contain errors, these may be amplified rather than corrected. We encourage future research to explore mechanisms for auditing verifier reliability, reducing the energy footprint of RL training, and ensuring equitable performance across diverse application domains.

---

**Case Study on TATTOO**

**(i) Before RL Training**

Okay, I need to verify step by step for the provided answer and determine if each step is correct or not.

**[Step 0 Evaluation]**
For step 0, let me first check whether it retrieves the correct information from the input table ... ...
**[Step 3 Evaluation]**
Looking at step 3 and the table data provided, I need to verify whether the summarization calculation on the column "capacity in persons / hour" is correct. Let me add each part together:

$350 + 500 + 970 + 10000 + 2000 + 2800 + 650 + 650 + 1000 + 1000 + 720 + 820 + 1000 = 16{,}920$
*(wrong PRM calculation)*

Since the provided step 3 gives a final calculation result of 22,460, which is not equal to 16,920. Step 3 is incorrect due to the wrong summarization on the overall carrying capacity.

The step is `\boxed{-1(incorrect)}` (✗ *Wrong Step Reward*)

---

**(ii) After RL Training**

Alright, I am given a complete table, a question, and a corresponding answer. I need to check the correctness of each step of the answer. Let me verify step by step:

**[Step 0 Evaluation]**
I start by checking if Step 0 pulls the proper information from the table ...
...
**[Step 3 Evaluation]**
To verify Step 3, I need to check the summarization (sum) over the "capacity in persons / hour" column. I will write Python code to verify this using the input table.

**Code Sandbox**

```python
table = {
    "columns": ["name or route", "capacity in persons / hour"],
    "data": [
        ["betten betten village", 350],
        ["betten village bettmeralp", 500],
        # ...
        ["trainerlift", 820],
        ["lager 1", 1000],
        # ...
    ],
}
capacity_index = table["columns"].index("capacity in persons / hour")
total_capacity = sum(row[capacity_index] for row in table["data"])

print("Total capacity in persons/hour: ", total_capacity)
```

**Code Interpreter**

```
Total capacity in persons/hour:  22460
```

Since the computed total of 22,460 aligns with the result in step 3, this step is correct.

The step is `\boxed{1(correct)}` (✓ *Correct Step Reward*)

Figure 9: Case study on TATTOO before and after RL training. Applying RL training enhances our PRM's dynamic utilization of tool calls, which in turn provides more reliable supervision over the input reasoning trajectories of LRMs.

