# OpenReview forum: "TaTToo: Tool-Grounded Thinking PRM for Test-Time Scaling in Tabular Reasoning"
_ICLR.cc/2026/Conference — ICLR 2026 Poster_

### Official Review · Reviewer_ysau · 2025-10-31

**Soundness:** 3
**Presentation:** 3
**Contribution:** 3
**Rating:** 6
**Confidence:** 3

**Summary:**

This paper introduces TaTToo (Tool-Grounded Thinking PRM), a Process Reward Model (PRM) framework specifically designed for Test-Time Scaling (TTS) in the domain of tabular reasoning. The authors empirically demonstrate that existing PRMs, which are widely adopted for supervising text-only reasoning steps, struggle with table-specific operations such as sub-table retrieval and schema interaction. To address this, TaTToo proposes a novel Tool-Grounded thinking process to finely supervise Large Reasoning Models (LRMs) during inference on structured data. The work leverages Reinforcement Learning (RL) and Reward Shaping to enhance the PRM's ability to capture fine-grained tabular reasoning signals, highlighting the significant potential of PRMs in structured reasoning domains like fact-checking, scientific analysis, and decision support.

**Strengths:**

S1: The paper is highly original, successfully extending the advanced PRM paradigm to the complex, structured domain of tabular reasoning. It precisely identifies the core limitation—the inability of current PRMs to handle table-specific operations—and provides a critical solution with the TaTToo framework. This is a vital step toward improving the reliability of LLMs on structured data.
S2: The methodology is sophisticated, employing Reinforcement Learning and Reward Shaping to effectively capture and optimize fine-grained signals of the reasoning process, demonstrating high-quality research execution.
S3: The paper is clear in articulating the problem, the proposed method, and its potential limitations. Method details (e.g., data pipeline) are well-described.

**Weaknesses:**

W1: The paper acknowledges the significant computational overhead from the RL stage, which may "hinder reproducibility and accessibility in low-resource environments." The authors should provide a deeper discussion and specific efficiency analysis in the main body (e.g., training time/cost comparison to SFT-Only models).
W2: Shallow comparison with recent tabular reasoning RL methods (e.g., Table-R1 series), only mentioned as related work.
W3: A critical risk, as highlighted in the ethics section, is that reliance on automated verification could amplify errors if the underlying tools or training data are flawed. The authors should include experiments to measure TaTToo's robustness against tool errors (e.g., errors in SQL compilers/executors) or noisy training data, and propose concrete, preliminary mechanisms for auditing verifier reliability.

**Questions:**

Q1:How do tool execution errors (e.g., code runtime exceptions) impact PRM reward supervision? The paper integrates tool outputs into rationales but lacks error handling mechanisms.
​​Q2:The paper notes schema interaction steps fail due to locality bias, and TATTOO uses table prefixes as a fix. Does this scale to very long reasoning chains (e.g., >10 steps)? Are there plans for dynamic prefix mechanisms?

---

> ### Author Response · Authors · 2025-11-21
> **Author Rebuttal (Part 1/3)**
>
> **Dear Reviewer ysau,**
>
> Thank you very much for your detailed feedback and for acknowledging the strengths of our work! We have thoroughly revised the paper in light of your comments and added several complementary experiments and analyses.  **In the updated paper, all modifications are marked in blue for easy reference.**
>
> Below, we provide point-by-point responses to address each of your questions and concerns. **For better clarity, we also include concise TL;DR before the extended explanations.**
>
> ---
> ## **W.1 Deeper Discussion & Effeciency Analyses on TaTToo**
>
> ## TL;DR
> Thank you for asking this insightful question! Below, we first (i) elaborate on how our current TaTToo design helps reduce training and inference costs, and (ii) provide a deeper efficiency analysis on TaTToo.
>
> ## Response
> **(i) Current Design for Efficient Training and Inference**
>
> - **Concise training data curation.** As described in `Section 4.2`, we progressively filter out low-quality samples across three stages, resulting in a high-quality 60K training set, which is **much smaller than typical PRM training corpora** (e.g., Qwen-PRM model uses ~800K samples [1, 2]). With this compact dataset and a lightweight backbone, TaTToo can be fully trained in **under 15 GPU hours**.
> - **Train once for general tabular reasoning.** Once trained, TaTToo can supervise step-level reasoning across a broad range of downstream tabular tasks and integrate seamlessly with multiple TTS strategies (e.g., Best-of-N, Beam Search). This shows that **TaTToo is not tied to a single dataset or use case and generalizes broadly across diverse tabular tasks.**
>
>
> **(ii) New Detailed Effeciency Analysis**
>
> We follow the reviewer’s suggestion and provide **an additional efficiency analysis** of TaTToo’s training overhead. Below is a detailed cost breakdown of our 8B model across **both the SFT and RL stages**, with total GPU cost estimated at **$4.8 per 8-GPU hours**.
>
> |TaTToo Training Stage|Training Data Size|# of Training Step|GPU setup|Total Hours|Total Cost|
> |-|-|-|-|-|-|
> |SFT|50K|~2400|8xA100|5.4|$25.9|
> |RL|10K|~280|8xA100|8.3|$39.8|
>
> **Key Findings.** The RL stage uses **~1/5 of the data** and **~1/9 of the training steps** compared to SFT, while it incurs higher compute due to rollout sampling and tool executions. Even so, the RL stage remains **short in duration** and rollout lengths are **modest**. Given the **large improvements over the SFT-only baseline** (see `Table 3`), we view this added compute as a **reasonable efficiency–effectiveness tradeoff** for the gains achieved by TaTToo.
>
> *Kind Note: We also added detailed efficiency discussion in our revised paper `Appendix M` ([click here](https://openreview.net/pdf?id=zc1ezBrr5m#page=28.10)) and will ensure later incorporate into the main body.*
>
> **References:**\
> [1] The Lessons of Developing Process Reward Models in Mathematical Reasoning.\
> [2] Let's verify step by step.
>
> ---
> ## **W.2 Comparison with Table-R1**
>
> ## TL;DR
> Thank you for pointing out the related work on RL-based method! As the response, we (i) first **clarify the relationship** between TaTToo and the mentioned RL methods (e.g., Table-R1 [1]), and then (ii) **include an additional experiment** comparing our approach with Table-R1.
>
> ## Response
>
> **(i) Relationship with Table-R1 series methods**
>
> TaTToo functions as a **reward model** offering **step-level supervision**, while the Table-R1 series are RL-trained **policy models**. As they target different components of the reasoning process, the methods are **complementary rather than directly comparable**, and TaTToo can be integrated with Table-R1–style policies during test-time scaling.
>
> **(ii) New experiments to integrate TaTToo on Table-R1**
>
> We further conduct an additional experiment comparing the performance of using Table-R1 alone versus integrating TaTToo with Table-R1, to demonstrate TaTToo’s reward supervision capabilities on such table-specialist policies.
>
> **Experiment Setups:**\
> We utilize the Table-R1-Zero (7B) from [1] and follow our experiment setting with the Best-of-N TTS strategy to evaluate across all 5 tasks.
>
> **Evaluation Results:**
> |Method |TB-NR|TB-FC|TB-DA|WTQ|MMQA|
> |-|-|-|-|-|-|
> |Table-R1-Zero|34.8|61.6|16.4|77.3|24.2|
> |Table-R1-Zero *with* TaTToo (Best-of-4) |39.6|64.7|18.3|80.9|26.8|
> |Table-R1-Zero *with* TaTToo (Best-of-8) |45.1|69.0|20.1|82.6|28.4|
> |Table-R1-Zero *with* TaTToo (Best-of-16)|48.2|72.3|23.5|84.5|30.3|
>
>
> **Key Findings.** We observe that integrating TaTToo with Table-R1-Zero consistently improves performance under the Best-of-N TTS setting as N increases. Similar to our experiments in `Table 1`, TaTToo continues to scale up Table-R1’s performance during inference without the bottlenecks seen in prior PRMs. **This demonstrates that TaTToo can serve as an effective step-level reward supervisor for RL-trained tabular reasoning policy models as well.**
>
>
> **Reference:**
>
> [1] Table-R1: Inference-Time Scaling for Table Reasoning.

---

> ### Author Response · Authors · 2025-11-21
> **Author Rebuttal (Part 2/3)**
>
> ## **W.3 & Q.1 TaTToo's Robustness against Tool Errors & Error Handling Mechanism**
>
> ## TL;DR
> We thank the reviewer for the detailed questions regarding tool execution errors and TaTToo's error handling mechanisms. As a response:
> - We first explain **how TaTToo currently handles tool execution errors**.
> - We then **provide new experiments and analyses** demonstrating TaTToo’s empirical robustness under erroneous tool-execution scenarios.
> - Finally, we **discuss additional error-mitigation mechanisms** that could further strengthen our tool-grounded PRM design.
>
> ## Response
> **(i) How TaTToo handles tool-execution errors currently**
>
> Based on our empirical observations during method design, there are mainly three types of tool-execution errors occurring during TaTToo's supervision process:
>
> - **Execution Errors:** Failures arising from the tool or execution environment itself (e.g., Python/Polars exceptions, environment issues, unsupported operations)
> - **Format Errors:** Tool errors caused by incorrect schema, invalid indices, or incorrectly structured inputs.
> - **LRM-source Errors:** Failures caused by incorrect CoT steps generated by the supervised LRM (e.g., incorrect or incomplete table contents, wrong reasoning logic).
>
> For all these three types of errors, we handle them through data-curation, training, and inference stages separately:
>
> **1. Training Data Curation**\
> During tool-use rationale synthesis, we adopt a **retry-and-recovery strategy**: when a tool call fails, we automatically **re-invoke and re-run the tool with the same input instruction**, up to a small retry limit (3 attempts by default), ensuring that collected tool outputs are executable and stable. Instances that still fail after the retry limit are excluded from the final curated dataset.
>
> **2. Dual-Stage Training**\
> As we stated in `Equation 3`, the tool-grounding term **support($\cdot$)** checks whether a rationale correctly incorporates *valid, executable* tool outputs. During training, if a tool call fails, we set **support($\cdot$) = 0**, so the step receives **no tool-grounding reward**. In addition, we **feed the returned error message back into the rationale**, allowing the PRM to learn explicit feedback from tool-execution failures. These mechanisms **prevent TaTToo from reinforcing faulty tool-grounded reasoning** and train the model to **down-weight or penalize steps associated with tool failures.**
>
> **3. Inference-time Scaling**\
> During inference, TaTToo adopts the same retry-and-recovery strategy used in data curation: each tool call is **re-executed up to 3 times** when an error occurs. This lightweight revocation mechanism improves the tool-execution pass rate and ensures that TaTToo delivers reliable reward supervision
>
> **(ii) Empirical Robustness under Imperfect Tool Execution**
>
> To further empirically demonstrate TaTToo’s robustness under erroneous tool-execution cases, we perform several empirical analyses on three tasks below.
>
> **[Analyses 1] Tool-execution error occurance ratio.**
>
> We first **quantify how frequently tool-execution failures occur during TaTToo’s verification process at inference**. We define a tool-execution error as any tool invocation that raises an exception or returns an invalid result.
>
> | Tool-Error Occurance Ratio | TB-NR | TB-FC |TB-DA|
> |-|-|-|-|
> |TaTToo|5.8%|1.9%|4.8%|
>
> We observe that tool errors occur at a very low rate across all tasks. This indicates that, **after our dual-stage tool-grounded training, TaTToo reliably generates executable tool calls and successfully integrates tools during inference** in the vast majority of cases.
>
> **[Analyses 2] How do tool execution errors impact TaTToo's reward supervision?**
>
> Building on Analysis 1, we further **examine whether TaTToo can still assign the correct reward when a tool execution fails during verification.** We define *"success rate under tool-error"* as correctly predicting the gold reward label despite the failed tool invocation.
> | Success Rate Under Tool-Error| TB-NR | TB-FC |TB-DA|
> |-|-|-|-|
> |TaTToo|73.6%|88.5%|78.5%|
>
> The result suggests that **TaTToo remains relatively robust under tool execution errors**, it correctly assigns rewards in 74–89% of cases across tasks. This demonstrates that **TaTToo has learned stable reasoning and reward patterns, allowing it to still provide reliable reward supervision when it cannot rely on successful tool execution.**
>
> **(iii) Additional Error-Handling mechanisms**
>
> In response to the reviewer’s suggestions, we outline several extensions that can further improve TaTToo’s robustness:
> - **Adaptive Retry Strategy:** Dynamically increase retry attempts based on error type or model confidence.
> - **Tool-Aware Step Refinement:** Automatically rewrite erroneous CoT steps using tool feedback before re-execution.
>
> We thank the reviewer for these suggestions and will explore these extensions to further strengthen TaTToo’s reliability under erroneous tool-execution conditions.

---

> ### Author Response · Authors · 2025-11-21
> **Author Rebuttal (Part 3/3)**
>
> ## **Q.2 Table Prefixing for Long Reasoning Chains**
>
> ## TL;DR
> Thank you for the insightful question regarding table prefixes and the scalability of our method! Below, we first clarify how our current method scales to long reasoning chains, and then discuss potential extensions toward more dynamic prefix mechanisms.
>
> ## Response
>
> **(i) Current Dynamic Table-prefix Assignment in TaTToo**
>
> >Does this scale to very long reasoning chains (e.g., >10 steps)?
>
> **Yes. In our implementation, table prefixes are only attached to schema-interaction steps, not to all CoT steps, which allows TaTToo to naturally scale to longer trajectories.** We explain in detail below.
>
> **1. Challenge in the Preliminary Study.** As discussed in `Line 221–223`, directly applying the table prefix is challenging because existing PRMs cannot automatically identify schema-interaction steps, and naively prefixing to all steps can easily exceed models' context length due to extremely long input.
>
> **2. Corresponding Solution on TaTToo.** The challenge above is **exactly why** we design TaTToo with dynamic prefix assingment:
>
> - **Data curation (`Line 265-266`):** We prepend the accurately retrieved sub-table as a prefix for **schema-interaction steps only instead of all CoT steps**. This helps mitigate the locality bias issue regardless of the schema-interaction steps' position in a long chain.
> - **SFT training: (`Line 310-311`):** When we train TaTToo to learn the verification patterns on our constructed data, the model is optimized to incorporate the table prefix specifically for schema-interaction steps. **In other words, TaTToo is trained under a regime where prefixes appear only on schema-interaction steps, not across the entire CoT.**
> - **Inference:** During inference, TaTToo first automatically retrieves the accurate sub-region and then self-assigns it as the prefix before each schema-interaction step. Other steps are evaluated without table prefixes.
>
> **3. Empirical Evidence on Long Chains (>10 steps) in Current Experiments.** To further demonstrate that TaTToo operates stably on long CoT sequences, **we report the average number of steps TaTToo supervises in our existing experiments (`Section 5`)**.
>
> |Avg. # of Steps|TB-NR|TB-FC|TB-DA|WTQ|MMQA|
> |-|-|-|-|-|-|
> |TaTToo|46.2| 23.5|36.8|29.2|54.7|
>
> These results indicate that TaTToo’s prefix mechanism remains effective across long reasoning trajectories, not just short sequences.
>
>
> **(ii) Toward More Dynamic Prefix Mechanisms**
> >  Are there plans for dynamic prefix mechanisms?
>
>
>
>
> We also agree that more dynamic handling of table context is a promising direction for further improving scalability. We see several natural extensions that can be integrated into the TaTToo framework:
>
> - **Summarized table prefixes.** Instead of prepending the full sub-region as a prefix, we can prepend **compressed summaries** (e.g., group statistics, key–value sketches, or schema-level summaries) that retain the key signal needed for verification while reducing token usage.
> - **Tool-mediated context access.** Rather than always prepending a prefix, TaTToo could be extended to **invoke a table-lookup tool on demand** during verification, retrieving only the cells needed for a particular schema-interaction step.
>
> **We thank the reviewer for the insightful question and will continue exploring ways to further scale TaTToo to more complex and longer reasoning chains.**
>
> ---
> ## Happy to have further discussion!
> **We sincerely thank the reviewer again for your thoughtful review to help us enhance the paper quality! We hope our responses address your concerns, and we are happy to discuss if you have any further questions!**

---

> ### Author Response · Authors · 2025-11-27
> **Appreciation and Looking Forward to Your Feedback!**
>
> Dear Reviewer ysau,
>
> We sincerely thank you for your thoughtful and constructive review, which has greatly helped us improve our manuscript! We have carefully addressed each of your questions and concerns in our responses with new experiments and additional in-depth analyses of our method. As the discussion period is nearing its end, we would appreciate your feedback on whether our responses have addressed your concerns. Thank you so much for your time and consideration!
>
> Warm regards,
>
> Authors of TaTToo

---

### Official Review · Reviewer_xVtW · 2025-11-01

**Soundness:** 4
**Presentation:** 4
**Contribution:** 4
**Rating:** 8
**Confidence:** 4

**Summary:**

This paper addresses a critical issue that existing Process Reward Models (PRMs) struggle with table-specific operations and result in performance bottelnecks in tabular reasoning. To this end, this paper proposes TaTToo, a novel table-grounded PRM framework that provides precise reward supervision in reasoning steps. Valuable contributions contain the large-scale high-quality step-level annotations and tool-grounded dual-stage training paradigm involving table-aware supervised fine-tuning and RL training with step-level reward shaping. Experimental analysis demonstrates the effectiveness and generalizability of TaTToo.

**Strengths:**

1. This paper is very well organized and presented, integrating clear and logical descriptions of motivations, pre-analysis, theoretical analysis and experimental validation. I am very impressed to read this manuscript and learn much from it.

2. The motivation of addressing step-level supervision in tabular reasoning is critical and impressive as a very novel contribution, as the limitations of previous LRMs have been demonstrated in the pre-analysis section.

3. The design of the RL reward integrates multiple reasonable components, and the policy improvement has been validated by robust theoretical analysis.

4. Experimental results clearly show the effectiveness and generalizability of TaTToo, revealing the contributions of the key modules of TaTToo.

**Weaknesses:**

1. TaTToo is trained based on Qwen3-8B and outperforms previous verifers with larger model sizes. I am curious about the result of the setting that utilizes the raw Qwen3-8B-model without further training, yet this setting is missing.

2. As TaTToo integrates large-scale dual-stage training within TTS scenarios, the computational overhead may be significant. Is there some possible solutions to further improve the efficiency?

**Questions:**

See the weaknesses part.

---

> ### Author Response · Authors · 2025-11-21
> **Author Rebuttal**
>
> **Dear Reviewer xVtW,**
>
> Thank you very much for your insightful feedback and for acknowledging the strengths of our work! We have thoroughly revised the paper in light of your comments and added several complementary experiments and analyses.  **In the updated paper, all modifications are marked in blue for easy reference.**
>
> Below, we provide detailed, point-by-point responses addressing each of your questions and concerns. **For more convenient reading, we also include concise TL;DR summaries before the extended explanations whenever appropriate.**
>
> ---
> ## **W.1 Comparison of TaTToo with Raw Qwen3-8B**
> ## TL;DR
> Thank you for the insightful suggestion! We follow your suggestion to conduct **additional experiments to compare TaTToo with the raw Qwen3-8B model**.
>
> ## Response
> **Experiment Setting:** We directly prompt the raw Qwen3-8B model to produce step-level supervision, including both the reasoning rationale and a binary reward (“correct” / “incorrect”). The full prompt template used is shown below:
>
> ```markdown=
> You are a step-level verifier. Given the table, question, and a solution with multiple reasoning steps,
> evaluate each step independently. For every step, determine whether it is correct/incorrect and briefly justify your decision.
>
> [Question]
> {question}
>
> [Table]
> {table_content}
>
> [Reasoning Steps]
> {steps}
>
> Please respond in the following format for each step:
>
> Step {i}
> Rationale: <your explanation>
> Label: <correct/incorrect>
> ```
> We leave other settings the same as the experiments in our paper.
>
> **Evaluation Results:**
> |Method|TB-NR@4|TB-NR@8|TB-NR@16|TB-FC@4|TB-FC@8|TB-FC@16|TB-DA@4|TB-DA@8|TB-DA@16|
> |-|-|-|-|-|-|-|-|-|-|
> |Qwen3-8B|66.5|67.3|67.1|75.9|76.5|77.0|22.6|24.8|26.2|
> |**TaTToo**|**71.2**|**74.2**|**76.4**|**77.4**|**79.6**|**81.2**|**27.7**|**31.9**|**33.6**|
>
> **Key Findings.** TaTToo consistently outperforms the raw Qwen3-8B model across downstream tasks, demonstrating the value of training on our curated step-level tabular reasoning dataset. This underscores the importance of
> - **high-quality dataset construction** that exposes the model to tabular reasoning patterns and step-level reward signals;
> - **dual-stage training pipeline**, which enables the model to effectively leverage tool feedback for reliable reward supervision.
>
> ---
> ## **W.2 Efficiency on TaTToo Training**
>
> ## TL;DR
> We thank the reviewer for asking the important question regarding efficiency. Below, we will **first discuss the current design in TaTToo** that helps reduce training and inference costs, and **then present several possible extensions** that could further improve the efficiency of TaTToo.
>
> ## Response
> **(i) Current Design for Efficient Training and Inference**
>
> - **Concise training data curation.** As highlighted in `Section 4.2`, we progressively filter out low-quality and irrelevant samples across three data curation stages, yielding a concise and high-quality training set of 60K examples. **This overall data size is relatively small compared with typical PRM training corpora.** For instance, one of the baseline Qwen-PRM models we compare against was trained on roughly 800K samples [1, 2]. With this compact dataset and a relatively lightweight backbone, we can complete the entire TaTToo training process in **under 8 GPU hours**.
> - **Train once for general tabular reasoning.** Once TaTToo is trained, the resulting PRM can provide step-level supervision across a broad range of tabular reasoning tasks (e.g., question answering, fact-checking, data analysis) and can be paired with multiple TTS strategies (e.g., Best-of-N, Beam Search). This indicates that **TaTToo is not limited to a single dataset or using scenarios, but can be broadly applied across diverse tabular tasks.**
>
> **(ii) Additional Solutions to Improve Efficiency**
>
> In addition, we want to propose **several additional extensions** that can be integrated with TaTToo to further improve efficiency.
>
> - **From Training perspective:** Since the goal of the first SFT stage is primarily to teach the backbone model table-aware verification, we can potentially leverage existing trained generative PRM checkpoints and apply **additional curriculum learning using only the second RL stage**.
> - **From Data Perspective:** After SFT, we can select **only the most informative or high-disagreement samples for the RL stage**, reducing the effective training set size without compromising downstream performance.
> - **From Model Perspective:** We can explore **parameter-efficient finetuning methods** such as LoRA to reduce GPU memory usage. In addition, using **a more lightweight or quantized model as the PRM backbone** could further lower training costs.
>
> **References:**\
> [1] The Lessons of Developing Process Reward Models in Mathematical Reasoning.\
> [2] Let's verify step by step.
>
> ---
> ## Happy to have further discussion!
> **Thank you again for the thoughtful review. We hope our responses address your concerns and are happy to discuss any further questions!**

---

> > ### Comment · Reviewer_xVtW · 2025-11-21
> > **Thanks for your rebuttal!**
> >
> > I appreciate the authors' feedback that addresses all of my concerns, i.e., the effectiveness of TaTToo compared to the raw Qwen3 model, and the efficiency explanation and discussion about this framework. Currently, I am fully convinced that this paper is of high quality. On this basis, I will keep my current rating and raise my confidence.

---

> > > ### Author Response · Authors · 2025-11-22
> > > **Thanks for the Positive Rating!**
> > >
> > > Dear Reviewer xVtW,
> > >
> > > Thank you very much for the positive rating and thoughtful follow-up feedback! We are glad that our clarifications fully addressed your questions, and we sincerely appreciate your time, effort, and support of our work!
> > >
> > > Warm regards,
> > >
> > > The Authors of TaTToo

---

### Official Review · Reviewer_sq6W · 2025-11-02

**Soundness:** 3
**Presentation:** 3
**Contribution:** 3
**Rating:** 4
**Confidence:** 3

**Summary:**

This paper proposes a process reward model (PRM) named TATTOO to address the common failure modes in current RMs, like subtable retrieval, schema/column interaction, and so on. The authors build a dataset of 60k+ step-level annotations and design a two-stage training pipeline (SFT+RL) for the better PRM. Specifically, they propose a GRPO variant with label matching, confidence calibration, tool grounding as shaping signals to optimize stepwise validation and scoring. Experiments spanning five tasks show that TATTOO yields monotonic, non-saturating gains as N increases under Best-of-N, beam search, and DVTS, and surpasses strong baselines like Qwen2.5-Math-PRM-72B and GenPRM-32B.

**Strengths:**

1. Re-diagnoses PRM failure from table-specific steps, clearly separating error sources into retrieval and schema interaction. The “table prefix” study shows existing PRMs’ insensitivity to distant retrieval context, motivating table-aware rewards + tool execution for stepwise verification.
2. End-to-end data pipeline: from expert LRM trajectory pools and dual validation filters to replacing subjective heuristics with executable tools for verification.
3. Introduces tool-supported reward shaping (label matching, confidence calibration, tool grounding) that explicitly rewards using tool outputs inside verification rationales—distinct from prior PRMs that rely on purely textual discrimination/generation.

**Weaknesses:**

The main weaknesses are in experiment settings. Addressing these would substantially strengthen the paper.
1. The competitive baseline about outcome reward model is absent. And I wonder how an outcome reward model trained on your training samples (which are surely a solid, data-aspect contribution) performs.
2. The policy model in experiments is Distilled-Qwen-14B while the reward model is trained from Qwen3-8B (weak policy model + strong reward model). And I wonder if TATTOO still works on strong policy models such as Qwen3-series (like Qwen3-32B, Qwen3-30B-A3B) or gpt-oss.

**Questions:**

Please see the weaknesses above

---

> ### Author Response · Authors · 2025-11-21
> **Author Rebuttal (Part 1/2)**
>
> **Dear Reviewer sq6W,**
>
> Thank you very much for your insightful feedback and for acknowledging the strengths of our work! We have thoroughly revised the paper in light of your comments and added several complementary experiments and analyses. **In the updated paper, all modifications are marked in blue for easy reference.**
>
> Below, we provide detailed, point-by-point responses addressing each of your questions and concerns. **For your more convenient reading, we also include concise TL;DR summaries before the extended explanations whenever appropriate.**
>
> ---
> ## **W.1 Comparison with ORM Baselines**
>
> ## TL;DR
> Thank you for acknowledging our contribution to training data curation and for raising the insightful question regarding the ORM comparison. Below, we provide new experiments to **compare TaTToo with several output reward model (ORM) baselines**.
>
> ## Response
>
> **Experiment Setups:**
> We detail the training and implementation settings for the ORM baselines below. For the policy model and all additional configurations, we follow exactly the same setup as outlined in our paper.
>
> - **Generative ORM:** We follow [1] and our paper settings to apply dual-stage training (SFT + RL) on the same backbone model, Qwen3-8B, using our curated training data. In this setting, the model is trained to **generate rationales and then only produce a final output-level reward token (“correct” or “incorrect”)** for each complete instance.
> - **Discriminative ORM:** We follow [2,3] to directly train a classification head paired with the backbone Qwen3-8B to **classify whether a candidate answer's final solution is correct or incorrect without generating rationales.**
>
> **Evaluation Results:**
> |Method (*Best-of-16*)|TB-NR|TB-FC|TB-DA|WTQ|MMQA|
> |-|-|-|-|-|-|
> |Discriminative ORM|66.4|72.0|26.8|68.1|25.3|
> |Generative ORM|70.6|75.9|28.5|69.2|26.6|
> |**TaTToo**|**76.4**|**81.2**|**33.6**|**73.5**|**29.1**|
>
> **Key Findings.** From the experiment results, we observe that:
> - TaTToo empirically offers stronger supervision than the two ORM baselines. This indicates that **TaTToo’s process-level supervision provides denser and richer reward modeling signals**, which in turn contributes positively to the downstream policy model’s performance.
> - When comparing the two ORM baselines, the Generative ORM consistently outperforms the Discriminative ORM, suggesting that adding generative rationales provides more informative supervision than binary correctness labels alone. As the Generative ORM leverages more information from our curated training set, this further **highlights the rich verification rationales and supervision signals** provided by our curated dataset.
>
> **Reference:** \
> [1] Generative Reward Models.\
> [2] V-STaR: Training Verifiers for Self-Taught Reasoners.\
> [3] Training verifiers to solve math word problems.

---

> ### Author Response · Authors · 2025-11-21
> **Author Rebuttal (Part 2/2)**
>
> ## **W.2 Evaluating TaTToo on Stronger Policy Models**
>
> ## TL;DR
> We thank the reviewer for this thoughtful question. As a reply, we (i) first briefly justify the reason for us to use Distilled-Qwen-14B as the policy model. (ii) Then, to demonstrate that TaTToo is also effective on stronger policy models, we follow the reviewer's suggestion and further **evaluate the effectiveness of TaTToo on three new policy models**.
>
> ## Response
> **(i) Explain our Choice of Distilled-Qwen-14B**
>
> According to [1], the DeepSeek-R1-Distill-Qwen-14B (Distilled-Qwen-14B) model is a strong LRM trained on **distilled large-scale, high-quality long-CoT data from DeepSeek-R1**. We adopt it because (a) it already demonstrates **strong step-wise reasoning ability** that transfers well to tabular reasoning, and (b) the 14B size is **relatively lightweight to deploy** compared to larger LRMs such as Qwen3-30B.
>
> In addition, **we fully agree with the reviewer that evaluating TaTToo on stronger and more updated LRMs would further strengthen our paper**. Therefore, we additionally include new experiments on these larger LRMs below.
>
> **(ii) New Evaluations of TaTToo on 3 Stronger Policy Models**
>
> **Experiment Setups:**
> - **Stronger Policy Models:** We adopt three strong downstream policy models paired with TaTToo, including **Qwen3-32B (thinking mode)**, **Qwen3-30B-A3B (thinking mode)**, and **gpt-oss-20b**.
> - **Evaluation Tasks:** We follow the same Best-of-N TTS setting mentioned in `Section 5` of our submission and evaluate the three models on TB-DA, WTQ, and MMQA tasks.
> - **Baselines:** We compare with three strong step-verifier baselines including Skywork-PRM-7B, GenPRM-32B, and Qwen2.5-Math-PRM-72B.
>
> **Evaluation Results:**
>
> - **Qwen3-32B Results**
>
> |Verifier (Best-of-N)|Params|TB-DA@4|TB-DA @8|TB-DA @16|TB-DA @32|WTQ@4|WTQ@8|WTQ@16|WTQ@32|MMQA@4|MMQA@8|MMQA@16|MMQA@32|
> |-|-|-|-|-|-|-|-|-|-|-|-|-|-|
> |Skywork-PRM-7B|7B|34.6|35.8|35.6|35.3|78.5|80.1|81.4|81.9|37.2|38.6|39.1|39.4|
> |GenPRM|32B|38.1|38.5|39.2|38.7|81.2|82.9|84.0|84.6|39.4|40.5|41.8|42.7|
> |Qwen2.5-Math-PRM-72B|72B|37.4|38.3|39.4|39.1|82.3|84.2|86.1|86.7|**42.2**|**43.8**|44.2|44.8|
> |**TaTToo**|**8B**|**38.3**|**39.5**|**40.5**|**41.3**|**83.8**|**86.5**|**87.3**|**88.6**|41.7|**43.8**|**44.7**|**46.3**|
>
>
> - **Qwen3-30B-A3B Results**
>
> |Verifier (Best-of-N)|Params|TB-DA@4|TB-DA @8|TB-DA @16|TB-DA @32|WTQ@4|WTQ@8|WTQ@16|WTQ@32|MMQA@4|MMQA@8|MMQA@16|MMQA@32|
> |-|-|-|-|-|-|-|-|-|-|-|-|-|-|
> |Skywork-PRM-7B|7B|29.4|31.1|31.9|32.2|70.2|72.0|73.9|75.0|29.4|30.9|32.0|33.1|
> |GenPRM|32B|30.9|33.7|34.2|33.9|71.6|73.4|75.8|77.3|30.7|32.5|33.6|35.0|
> |Qwen2.5-Math-PRM-72B|72B|**34.8**|35.3|35.6|37.2|72.5|74.6|77.1|78.9|31.9|33.8|35.0|36.5|
> |**TaTToo**|**8B**|34.4|**35.8**|**37.6**|**39.1**|**73.8**|**75.8**|**78.3**|**80.4**|**33.1**|**34.7**|**36.2**|**38.0**|
>
> - **gpt-oss-20b Results**
>
> |Verifier (Best-of-N)|Params|TB-DA@4|TB-DA @8|TB-DA @16|TB-DA @32|WTQ@4|WTQ@8|WTQ@16|WTQ@32|MMQA@4|MMQA@8|MMQA@16|MMQA@32|
> |-|-|-|-|-|-|-|-|-|-|-|-|-|-|
> |Skywork-PRM-7B|7B|27.2|28.3|27.9|28.1|71.0|72.8|74.2|74.5|33.4|38.1|38.5|38.8|
> |GenPRM|32B|29.0|30.2|31.1|31.0|73.5|75.6|77.8|78.0|37.2|39.3|39.6|40.1|
> |Qwen2.5-Math-PRM-72B|72B|**33.1**|**34.7**|34.9|35.5|74.2|76.5|78.7|79.3|**39.5**|40.5|43.4|43.9|
> |**TaTToo**|**8B**|32.8|34.4|**36.2**|**37.9**|**76.0**|**78.1**|**80.4**|**82.2**|39.2|**42.1**|**44.0**|**45.7**|
>
> **Key Findings.** Across all three stronger policy models, incorporating TaTToo consistently leads to better downstream performance compared with larger size baseline PRMs. **This aligns with our observations in `Section 5` and further demonstrates TaTToo’s supervision effectiveness across policy models of varying sizes.**
>
> *Kind Note: For more details, we also incorporated the new experiments above into our revised paper in `Appendix I` ([click here](https://openreview.net/pdf?id=zc1ezBrr5m#page=25.10))*
>
> **Reference:**\
> [1] DeepSeek-R1: Incentivizing Reasoning Capability in LLMs via Reinforcement Learning.
>
> ---
> ## Happy to have further discussion!
> **We sincerely thank the reviewer again for the thoughtful review to help us enhance the paper quality. We hope our responses address your concerns, and we are happy to discuss if you have any further questions!**

---

> ### Author Response · Authors · 2025-11-27
> **Appreciation and Looking Forward to Your Feedback!**
>
> Dear Reviewer sq6W,
>
> We sincerely thank you for your thoughtful and constructive review, which has greatly helped us improve our manuscript! In response to your concerns, we have conducted additional experiments and provided an in-depth analysis of our method. As the discussion period is nearing its end, we would appreciate your feedback on whether our responses have addressed your concerns. Thank you so much for your time and consideration!
>
> Warm regards,
>
> Authors of TaTToo

---

### Author Response · Authors · 2025-12-03
**To AC: Summary of Reviews, Paper Revision, and Rebuttal. Thank You for Your Efforts!**

Dear PCs, SACs, ACs, and Reviewers,

Thank you very much for your guidance and valuable contributions to our work. We also sincerely appreciate the reviewers’ thoughtful and constructive feedback, and we are encouraged by the positive assessments following our rebuttal.

In response to the insightful comments, **we have conducted substantial new experiments and in-depth analyses, and accordingly revised the manuscript ([click here](https://openreview.net/pdf?id=zc1ezBrr5m))** to improve both clarity and technical completeness. To assist the AC and help reduce review workload, we briefly summarize the key points raised by the reviewers and our corresponding updates below.

---
## **Strengths and Contributions**
**1. Novel and important problem setting for tabular TTS (Reviewers xVtW, ysau).** Reviewers highlighted that extending PRMs from text-only reasoning to table-specific operations addresses a critical and under-explored bottleneck in TTS for tabular reasoning.

**2. Well-designed tool-grounded PRM framework and dual-stage training (Reviewers sq6W, xVtW, ysau).** The proposed tool-grounded thinking process and dual-stage training pipeline were recognized as sophisticated and well-motivated, with clear methodology and theoretical underpinnings.

**3. High-quality step-level dataset and strong empirical results (Reviewers sq6W, xVtW, ysau).** The construction of 60k+ step-level annotations and evaluations on five tabular benchmarks was recognized as a valuable contribution, demonstrating monotonic TTS gains and consistent improvements over strong baselines.

**4. Clear presentation and detailed analysis (Reviewers xVtW, ysau).** Reviewers agreed the paper is well organized and clearly written, with logically structured motivation, pre-analysis, method description, and ablations that make the framework and its implications easy to follow.

---
## **How We Addressed the Reviewers’ Concerns**
**1. Adding Output Reward Model (ORM) Baselines (Reviewer sq6W)**
**Quick Link:** [Reply to sq6W – ORM Comparison](https://openreview.net/forum?id=zc1ezBrr5m&noteId=5PW8xb654y)
- We newly compared with **two ORM baselines**, including a discriminative ORM and a generative ORM.
- Under the same Best-of-N TTS setups, we showed that **TaTToo provides richer reward supervision and consistently outperforms both ORM variants** across all five tabular benchmarks. Together, these new experiments empirically support the advantage of TaTToo’s process reward modeling.

**2. Evaluating TaTToo on Stronger Policy Models (Reviewer sq6W)**
**Quick Link:** [Reply to sq6W – Strong Policy Models](https://openreview.net/forum?id=zc1ezBrr5m&noteId=hObNqatL9o)
- We integrated TaTToo on **three additional stronger policy models** (Qwen3-32B, Qwen3-30B-A3B, and gpt-oss-20b).
- Across all three strong policy models, TaTToo consistently improves downstream accuracy. These studies confirm that **TaTToo’s supervision transfers robustly to stronger LRMs**

**3. Robustness to Tool Errors (Reviewer ysau)**
**Quick Link:** [Reply to ysau – Robustness](https://openreview.net/forum?id=zc1ezBrr5m&noteId=ilLzD0cX9P)

- We explained our method's retry-and-filter error handling strategy and added new analyses showing TaTToo's **low tool error rates** with **high reward accuracy under failures**.
- We clarified that table prefixes are applied only to schema-interaction steps, enabling scalability to long reasoning chains without context explosion.

---

## **Reviewers’ Engagement Summary During Rebuttal**

We sincerely thank all reviewers for their positive feedback and active engagement throughout the discussion period. In summary:

**1. Reviewer xVtW** gave a **positive assessment of score 8 at start**. After reading our rebuttal, the reviewer explicitly stated they are **“fully convinced that this paper is of high quality”** and **increased the confidence to 5** on Nov 21.

**2. Reviewer ysau** offered a supportive and positive assessment **with a score of 6** from the start.

**3. Reviewer sq6W** has not yet had the opportunity to engage in the discussion. However, the reviewer assigned **positive subscores (3:good) for Soundness, Presentation, and Contribution**, and noted that the concern mainly lay in the experimental settings. **In our rebuttal response and revised paper, we have conducted new experiments with detailed analyses to address the two questions regarding experimental settings.**

---
Once again, we sincerely thank the reviewers, AC, SAC, and PC for their careful reading and constructive feedback. Your insightful comments have substantially strengthened both the technical rigor and the clarity of our paper! Thank you for your time and efforts!

Warm regards,

The Authors of TaTToo

---

### Meta-Review · Area_Chair_Nyaa · 2026-01-07

**Summary:**

This paper proposes TaTToo, a table-grounded Process Reward Model (PRM) designed to address well-known failure modes of existing PRMs in tabular reasoning, such as sub-table retrieval errors and schema/column interaction issues. The work makes a strong case that text-only PRMs are insufficient for structured data and introduces a tool-grounded, step-level supervision framework trained via a two-stage SFT + RL pipeline. The authors also contribute a sizable, carefully curated dataset of over 60k step-level annotations and introduce principled reward shaping signals (label matching, confidence calibration, and tool grounding). Across multiple tabular reasoning benchmarks and test-time scaling regimes, TaTToo demonstrates consistent, monotonic improvements with increasing N and outperforms strong existing PRMs, often despite being smaller in model size.

There is broad agreement that the paper is well-motivated, technically sound, and impactful. Reviewers particularly praise: 1. the clear diagnosis of PRM failure modes specific to tabular reasoning, supported by targeted analyses; 2. the end-to-end design spanning data curation, model training, and inference-time integration; 3. the novelty of explicitly grounding reward modeling in executable tools rather than relying solely on textual discrimination; and 4. strong empirical results showing robustness and scalability across tasks, inference strategies, and even stronger downstream policy models.

The main concerns raised centered on experimental completeness and practical considerations: missing comparisons to outcome reward models, the initial use of a weaker policy model paired with a stronger reward model, computational overhead from RL and tool usage, limited discussion of related RL-based tabular methods, and robustness to tool errors. The authors’ rebuttal is thorough and largely convincing. They add new ORM baselines, showing TaTToo’s clear advantage, evaluate TaTToo with substantially stronger policy models, include comparisons against raw Qwen3-8B, and provide detailed efficiency analyses with concrete GPU-hour and cost breakdowns. Additional experiments demonstrate robustness to tool execution errors and show that TaTToo complements RL-trained tabular policies such as Table-R1. These additions substantially strengthen the paper and directly address the reviewers’ questions.

Overall, the reviews and rebuttal indicate that this is a solid and timely contribution to process reward modeling and structured reasoning. The paper extends PRMs into tabular domains with a principled, tool-grounded approach and is supported by comprehensive empirical validation. While one reviewer initially rated it slightly below threshold, their concerns were largely resolved by the rebuttal, and the overall balance of evidence favors acceptance.

**Reviewer Concerns:**

see above

**Reviewer Scores:**

see above

---

### Decision · Program_Chairs · 2026-01-26

Accept (Poster)